

# Quantifying the range of the dust direct radiative effect due to source mineralogy uncertainty

Longlei Li[1], Natalie M. Mahowald[1], Ron L. Miller[2], Carlos Pérez García-Pando[3,9], Martina Klose[3], Douglas S. Hamilton[1], Maria Gonçalves Ageitos[3,10], Paul Ginoux[4], Yves Balkanski[5], Robert O. Green[6], Olga Kalashnikova[6], Jasper F. Kok[7], Vincenzo Obiso[2,3], David Paynter[8], David R. Thompson[6]

[1]Department of Earth and Atmospheric Sciences, Cornell University, Ithaca, NY, United States
[2]NASA Goddard Institute for Space Studies, New York, NY, United States
[3]Earth Sciences Department, Barcelona Supercomputing Center, Barcelona, Spain
[4]Atmospheric and Oceanic Sciences Program, Princeton University, Princeton, NJ, United States
[5]Laboratoire des Sciences du Climat et de l'Environnement, UMR 8212 CEA-CNRS-UVSQ-UPSaclay, Gif-sur-Yvette Cedex, France
[6]Jet Propulsion Laboratory, California Institute of Technology, Pasadena, CA, USA
[7]Atmospheric and Oceanic Sciences, University of California, Los Angeles, CA, United States
[8]Geophysical Fluid Dynamics Laboratory, Princeton, NJ, United States
[9]ICREA, Passeig Lluís Companys 23, 08010 Barcelona, Spain
[10]Department of Project and Construction Engineering, Technical University of Catalonia, Terrassa, Spain

*Correspondence to*: Longlei Li (ll859@cornell.edu)

**Abstract.** The large uncertainty in mineral dust direct radiative effect (DRE) hinders projections of future climate change due to anthropogenic activity. Resolving modelled dust mineral-speciation allows for spatially and temporally varying refractive indices consistent with dust aerosol composition. Here, for the first time, we quantify the range in dust DRE at the top of the atmosphere (TOA) due to current uncertainties in the surface soil mineralogical content using a dust mineral-resolving climate model. We propagate observed uncertainties in soil mineral abundances from two soil mineralogy atlases along with the optical properties of each mineral into the DRE and compare the resultant range with other sources of uncertainty across six climate models. The shortwave DRE responses region-specifically to the dust burden depending on the mineral speciation and underlying shortwave surface albedo; positively when the regionally averaged annual surface albedo is larger than 0.28, and negatively otherwise. Among all minerals examined, the shortwave TOA DRE and single scattering albedo at the 0.44-0.63 μm band are most sensitive to the fractional contribution of iron oxides to the total dust composition. The global net (shortwave plus longwave) TOA DRE is estimated to be within -0.23 to +0.35 W m$^{-2}$. Approximately 97% of this range relates to uncertainty in the soil abundance of iron oxides. Representing iron-oxide with solely hematite optical properties leads to an overestimation of shortwave DRE by +0.1 W m$^{-2}$ at the TOA, as goethite is not as absorbing as hematite in the shortwave spectrum range. Our study highlights the importance of iron oxides to the shortwave DRE: they have a disproportionally large impact on climate considering their small atmospheric mineral mass fractional burden (~2%). An improved description of iron oxides, such as those planned in the Earth Surface Mineral Dust Source Investigation (EMIT), is thus essential for more accurate estimates of the dust DRE.



## 1 Introduction

Mineral dust emitted from erodible land surfaces has myriad impacts on the Earth System and humanity society by perturbing the radiation budget (Tegen and Fung, 1994; Sokolik and Toon, 1996), interacting with cloud processes (Rosenfeld et al., 2001; DeMott et al., 2003; Mahowald and Kiehl, 2003; Atkinson et al., 2013), affecting ocean and land

biogeochemical cycles (Swap et al., 1992; Jickells et al., 2005; Mahowald et al., 2017), causing respiratory and cardiovascular disease (Meng and Lu, 2007), contributing to other ailments like meningitis (Pérez García-pando et al., 2014), and modifying atmospheric chemistry (Dentener et al., 1996; Martin et al., 2003). Dust aerosol (here defined as soil particles suspended in the atmosphere) perturbs the radiative energy balance directly by scattering and absorbing shortwave and longwave radiation and indirectly by changing the cloud albedo and lifetime by acting as cloud condensation nuclei (CCN)

and ice nuclei (IN) (Nenes et al., 2014). These two processes through which perturb the radiative energy balance are denoted as aerosol-radiation and aerosol-cloud interactions, respectively (Boucher et al., 2013). Through interactions with radiation and cloud, dust can feedback upon meteorology in the planetary boundary layer, the large-scale circulation, and the energy, water and carbon cycles (Miller and Tegen, 1999; Perlwitz et al., 2001; Pérez et al., 2006; Solmon et al., 2008; Lau et al., 2009; Mahowald et al., 2011; Shao et al., 2011).

At the global scale, mineral dust is estimated to warm the atmosphere and cool the Earth's surface in the shortwave spectral range, and induces opposite effects in the longwave spectral range (Sokolik and Toon, 1996; Kok et al., 2017). However, these estimates are currently highly uncertain. A recent review which synthesized data on dust abundance, optical properties and size distribution estimated that dust causes a net (shortwave plus longwave) direct radiative effect (DRE) ranging from -0.48 to +0.20 W m$^{-2}$ (Kok et al, 2017). This degree of uncertainty in the net DRE of dust constitutes an important gap in our

understanding of the role it plays in climate.

Much of the DRE uncertainty can be attributed to uncertainties in the dust aerosol composition and its evolution during transport (Hand et al., 2004; Baker and Croot, 2010; Shao et al., 2011). Most of the abovementioned impacts of dust aerosols on climate are closely related to the composition of minerals in dust particles: 1) the dust DRE in some longwave bands depends on quartz or calcite, and across many shortwave bands dust strongly depends on the iron oxides content and its

mixing state with other minerals (Sokolik et al., 1998; Sokolik and Toon, 1999); 2) chemical reactions occurring on the dust particle surface depend on dust minerals (particularly, calcite) and chemical composition (Dentener et al., 1996; Hanisch and Crowley, 2003; Kumar et al., 2014); 3) the liquid water uptake rate and ice nucleation ability of dust is determined by its hygroscopicity, size, and shape and thus related to the physio-chemical properties of the minerals (e.g., feldspar) (Karydis et al., 2011; Atkinson et al., 2013); 4) after atmospheric processing, iron-bearing minerals (e.g., hematite, goethite, illite, and

hydroxide) contained in dust aerosols contribute a large fraction of the atmospheric bioavailable iron flux to remote ocean regions. This can cause dust-iron fertilization to occur and thus influences ocean marine primary productivity and biomass accumulation (Meskhidze et al., 2003; Journet et al., 2008; Schroth et al., 2009; Hamilton et al., 2020); 5) phosphorus-bearing minerals are important for marine and terrestrial biogeochemistry effects, for example, the north Pacific Ocean and



Amazon rainforest (Swap et al., 1992; Okin et al., 2004; Letelier et al., 2019). Currently, the soil mineral composition

required by dust-speciated models are provided by either Claquin et al. (1999, C1999 hereafter) – with additional extrapolation to other soil types (three new soil units and soil phosphorous) proposed by Nickovic et al. (2012) – or Journet et al. (2014) (J2014 hereafter). The mineral composition of clay- (<2μm) and silt-sized (between 2 and 63 μm) particles is assumed to be related to the soil type in C1999, and the soil unit in J2014. Because of limited measurements, many of which are not located in major dust emission regions, global atlases of soil mineral distribution are based on extensive extrapolation

and thus have a large uncertainty (Claquin et al., 1999; Journet et al., 2014; Perlwitz et al., 2015a, 2015b; Scanza et al., 2015).

A technique to model dust aerosol optical properties, accounting for their physicochemical characteristics, was proposed by Sokolik and Toon (Sokolik and Toon, 1999). The authors demonstrated, via offline radiative transfer calculations, that the DRE by mineral dust was highly dependent on the representation of its mineral-specific absorption properties. They

suggested that internal mixing of iron oxides (hematite and goethite) with less absorptive minerals enhances the absorption of shortwave radiation and can reverse the sign from a negative (cooling) to positive (warming) DRE at the top of the atmosphere (TOA). Later studies (Alfaro et al., 2004; Lafon et al., 2006; Balkanski et al., 2007; Formenti et al., 2014; Li and Sokolik, 2018) confirmed the importance of iron oxides to the shortwave dust DRE, particularly near dust source areas, even when they are mixed with particles that are also strongly absorbing (e.g. black carbon) (Alfaro et al., 2004). Two main types

of iron oxide minerals are found in soils: hematite and goethite (Journet et al., 2014). Iron in both minerals is generally to be found in a (III) oxidation state, but they have distinct optical properties in the shortwave spectrum; hematite exhibits a more pronounced spectral absorption and has a comparatively stronger ability to absorb shortwave radiation than goethite. Consequently, the calculated estimates of the single scattering albedo (SSA) for hematite- and goethite-clay aggregates, with the same size distribution, are significantly different (Lafon et al., 2006). Iron oxides represent 2.4-4.5% of the total dust

mass (Formenti et al., 2008), although a slightly larger range (0.7-5.8%) of iron oxides in dust was reported in a more recent study (Di Biagio et al., 2019). North African samples exhibited a dominance of goethite over hematite (percentage mass content of iron oxides: 52-78% versus 22-48%, respectively (Formenti et al., 2014). The partitioning of these two iron oxides is thus necessary to accurately estimate the DRE, because of the difference in their optical properties and a strong regional variation in their soil content (Lafon et al., 2006; Formenti et al., 2014; Di Biagio et al., 2019).

Because of the importance of physio-chemical characteristics of different dust minerals to estimating the dust DRE at shortwave bands, one focus for dust model development is on improving the representation of dust minerals (Scanza et al., 2015; Perlwitz et al., 2015a) and their coupling with radiative transfer processes using mineral specific optical properties (Sokolik and Toon, 1999). Scanza et al. (2015) introduced eight minerals (illite, kaolinite, smectite, hematite, quartz, calcite, gypsum and feldspar) identified as climatically important by C1999 into the Community Atmosphere Model of version 4

(CAM4) and five minerals (illite, kaolinite, smectite, hematite, and a bulk remainder mineral) into version 5 (CAM5) based on C1999 (both CAM4 and CAM5 are embedded within the Community Earth System Model: CESM). Similarly, the eight



minerals within CAM4 were included in the NASA Goddard Institute for Space Studies (GISS) Earth System ModelE2 (Perlwitz et al., 2015a). These previous studies exhibited the models' limited ability to match the available observations of mineral fractions and ratios. This mismatch is attributed to the inherent limitations and uncertainties in the surface soil

mineralogy mapping (Perlwitz et al., 2015b; Scanza et al., 2015; Zhang et al., 2015b) along with uncertainties in the models' emission, transport, and deposition. Perlwitz et al. (2015a,b) and Pérez García-Pando et al. (2016) show that despite these uncertainties, reconstructing the emitted mineral aggregates from the disturbed soil mineralogy maps based upon brittle fragmentation theory (Kok, 2011) and additional empirical constraints better reproduces size-resolved mineralogy and elemental composition observations. Scanza et al. (2015) shows that CAM underestimates the observed DRE efficiency near

North Africa. This underestimate could be attributed to difficulty of DRE retrieval along with the large uncertainty in hematite in the C1999 soil mineralogy atlas, which includes a range of iron oxide abundance (0.0–7.0% by weight).

Here, for the first time, we undertake a detailed and systematic study of the sensitivity of the dust DRE resulting from current uncertainties in soil mineral composition. We compare the sensitivity of DRE to uncertainties in soil mineral composition to those from other sources, such as the range in measured complex refractive indices for dust minerals and dust

burdens. In this study we focus on composition of dust and do not examine other sources of uncertainty including the mineral vertical and size distributions, cloud processes, and surface albedo (Liao and Seinfeld, 1998; Li and Sokolik, 2018). In addition to C1999, as used in previous studies (Scanza et al., 2015; Perlwitz et al., 2015a), we incorporate results using the updated J2014 soil mineralogical atlas, which separates iron oxides into hematite and goethite. We focus on the sensitivity studies within only one model (CAM5), and then compare results to three other models, GISS ModelE2, Multiscale Online

Non-hydrostatic AtmospheRe CHemistry model (MONARCH; previously known as NMMB/BSC-CTM), and Geophysical Fluid Dynamics Laboratory (GFDL) (see Section 2.2 for descriptions) to examine both parametric and structural uncertainty sources.

## 2 Methods

### 2.1 Descriptions of soil mineralogy data

Two datasets currently exist that can be used to describe the size-resolved mineralogical composition for potential dust sources around the globe. For both datasets, the soil mineralogical composition was inferred based on the hypothesis that the surface mineralogy depends on the size distribution, and physio-chemical properties (e.g., appearance color) of the soil.

The first dataset was originally created by Claquin et al. (1999), who compiled measurements linking soil type and mineral

composition from the available literature. This dataset contains information regarding an average relative abundance of eight minerals (mean mineralogy table, MMT) in the clay-sized and silt-sized categories for 28 soil types that are considered wind erodible. Illite, kaolinite, and smectite (only present in the clay-sized category) frequently dominate over calcite and quartz among different soil types. In the silt size category, the dominant minerals are quartz and/or feldspar instead of hematite,



gypsum, and calcite, except for salt flats where calcite is dominant. Also included in C1999 is the standard deviation of the
mean mineral content for the 28 soil types. This study extends hematite to the clay size category by assigning the same
fraction as it is in the silt category and subtracting the same mass from illite consistent with recent studies (Balkanski et al.,
2007; Nickovic et al., 2012; Scanza et al., 2015; Perlwitz et al., 2015a). The global map of arid surface mineralogy is created
following Claquin et al. (1999) and Scanza et al. (2015) via the FAO/UNESCO WGB84 at 5'x5' arc minutes with soil legend
from FAO/UNESCO Soil Map of the World in 1976 (Batjes, 1997) using the MMT.


The other soil mineral dataset presented in Journet et al. (2014) (J2014) is an extension of C1999. It includes four additional
minerals, one (vermiculite) in the clay-sized soil category, two (mica and goethite) in the silt-sized category, and one
(chlorite) in both categories. The mean mineralogical content was assigned to different soil units, as classified by FAO
(FAO-UNESCO, 1974: 135 soil units; FAO, 1990: 193 soil units). The standard deviation is also provided but only for a
limited number of soil units. Compared to C1999, this more recent compilation is not confined to the soil units that are
located in arid and semi-arid areas, and benefits from a use of more extensive literature. Nevertheless, there is a number of
soil units lacking mineralogical information (the mean mineralogical content and in particular the associated standard
deviation), especially for the silt-sized soil class where the information is scarce. The mean mineralogical content for these
missing soil units was thus characterized through assumptions rather than observation-derived data. For iron oxides, which
are relevant to the DRE of dust, data are present for only 23% (~45) of the reported soil units. We fill soil units without the
mean mineralogy content including iron oxides with the mineralogical composition of the major soil unit they belong to. Our
mineralogy maps created according to this dataset rely on the dominant soil unit at 0.5º×0.5º resolution, as derived from the
Harmonized World Soil Database v1.21 (FAO/IIASA/ISRIC/ISSCAS/JRC, 2012) map at 30 arc seconds of horizontal
resolution. Mean mineralogy values are then geographically assigned according to the relevant soil units.



## 2.2 Model descriptions

Model sensitivity analysis in this paper focuses on results from the Community Earth System Model (CESM). To assess a spread in the sensitivity of DRE to representations of dust cycles, we compare CESM to three other models (GISS ModelE2, MONARCH, and GFDL), as described in this section. We employ three versions of the Community Atmosphere Model (CAM) in CESM following Scanza et al. (2015): the Bulk Aerosol Model (BAM) in the CAM4 (Neale et al., 2013), and the Modal Aerosol Model (MAM) in CAM5 (Hurrell et al., 2013) and CAM6 (Danabasoglu et al., 2020). In these CAM versions, the DRE is calculated by speciating dust into minerals (Section 2.2.1). We construct perturbation sensitivity analyses with CAM5 only (Section 2.3.1), as the DRE in CAM4 is insensitive to dust minerals (Section 3.2.2.1) and the high resolution CAM6 model is computationally expensive (a factor of 10 times more core hours are required in CAM6 compared to CAM5 (Hamilton et al., 2019), particularly considering the large number of simulations needed.

Mineral composition is also calculated using an updated version of the NASA ModelE2 (Schmidt et al. 2014) (Section 2.2.2) as described in Perlwitz et al. (2015a,b) and Pérez García-Pando et al. (2016). Since the relation of the DRE to simulated minerals in this model is still under development, we apply a statistical relationship between simulated minerals and shortwave dust DRE in CAM5 to predict the shortwave DRE (Section 2.3.4) based on simulated minerals in GISS ModelE2. The MONARCH (Section 2.2.3) and GFDL models (Section 2.2.4) does not include dust mineral speciation, so, we use the DRE related to bulk dust AOD (Section 2.3.4).

### 2.2.1 Community Earth System Model

Dust mineral speciation (illite, kaolinite, montmorillonite, hematite, quartz, calcite, feldspar, and gypsum) was incorporated for CAM4 (Scanza et al. 2015) and CAM5 (Scanza et al 2015; Hamilton et al 2019) using C1999. Here we add a new mineral tracer for goethite to CAM5 to use J2014 (Section 2.1) and adopt the incorporated CAM5 mineral species when using C1999. Recently, a new CAM6 model for CESM2 was released which was updated to an improved two-moment prognostic cloud microphysics, MG2, (Gettelman and Morrison, 2015) from MG (Morrison and Gettelman, 2008) used in CAM5. For this study, we incorporate the mineral speciation of CAM5, closely related to the Department of Energy model: Energy Exascale Earth System Model (E3SM) (Liu et al., 2016; Lauritzen et al., 2018; Caldwell et al., 2019), into the CAM6 model. Each mineral was emitted, transported and deposited separately in the model. Aerosols including dust in both CAM5 and CAM6 are subdivided into interstitial (within the clear air) and cloud-borne (within in clouds) particles for a better representation of advection and deposition processes, as documented in Liu et al. (2012). In the atmosphere each mineral individually interacts with the shortwave and longwave radiation.

The dust emission, transport, and deposition are simulated by the Dust Entrainment And Deposition model (DEAD, Zender et al., 2003) which has been implemented in the land and atmosphere components of  the CESM, and described in detail previously (Zender et al., 2003; Mahowald et al., 2006; Albani et al., 2015). The emission of dust occurs within non-





vegetated, dry soil regions, and is initiated once a friction velocity threshold has been exceeded. The threshold is a function of the soil state (e.g., soil moisture, snow cover, surface crust, vegetation cover) and near-surface meteorology (e.g., air density, horizontal wind speed). Vegetation tends to protect the soil from wind erosion by reducing the energy transfer of

wind momentum to the soil surface and is represented in the model via a linear dependence on the leaf area index (LAI) (Mahowald et al., 2006). No dust emission occurs within grid cells with a LAI exceeding 0.3 $m^2$ $m^{-2}$. The threshold wind speed for dust entrainment to the atmosphere increases with soil moisture following a semi-empirical relation between the threshold wind speed and soil moisture obtained by Fecan et al. (1999) with additional optimization from the traditional dependence of the square of clay mass fraction (Fecan et al., 1999; Zender et al., 2003).


The default dust model utilizes a prescribed soil erodibility source function (Ginoux et al., 2001) which associates dust emissions to topographical depressions where abundant erodible sediment accumulates (Ginoux et al., 2001; Zender et al., 2003; Mahowald et al., 2006). In this study, we used an updated physical dust emission scheme developed by Kok et al. (2014a), based on the brittle fragmentation theory (Kok, 2011) which has been shown to improve model-observation

comparisons without the source function (Kok et al., 2014b). The emitted size distribution, of either bulk dust (sum of all minerals or non-speciated dust) or minerals, is assumed to be independent of the soil properties of the source location and wind speeds (Albani et al., 2014; Perlwitz et al., 2015a; Scanza et al., 2015) and currently only considers the climatologically most relevant diameter range from 0.1-10 μm. Each mode in CAM5 or CAM6 represents the aerosol size distribution by a lognormal function with varying mode dry or wet particle radii. For CAM6, the default size distribution is to use a narrow

coarse mode width (geometric standard deviation: 1.2 compared to 1.8 in CAM5; Table 1) which does not adequately simulate size distribution of the dust aerosol mass. Thus, in the CAM6 simulations, we retained the mode size distribution of CAM5, which enables the use of the same fractional contributions of the clay- and silt-sized soil to the dust aerosol mass for the accumulation and coarse particle modes in CAM6 as in Scanza et al. (2015). The emission of each mineral into the Aitken mode in CAM5 and CAM6 are refined following that into the accumulation mode.


Dust mineral species carried within each mode in CAM5 and CAM6 are internally mixed with each other and with other non-dust species (e.g., sea salt, sulfate, black carbon, primary and/or secondary organic matter) in the same mode under the homogenous assumption (the same proportions of each components in any individual aerosol particle) but externally mixed between modes (Liu et al., 2012, 2016). In comparison, all aerosol species are externally mixed in CAM4, but the optical

properties for dust species (SSA, the extinction coefficient, and the asymmetry factor) are calculated offline using the MIEV0 software (Wiscombe, 1980) with a spherical shape assumption and prescribed aerosol size distribution independent of locations.

The radiative flux at each vertical model layer, at 19 and 14 shortwave bands (for CAM4 and CAM5/CAM6, respectively),

and 16 longwave bands, is computed by the rapid radiative transfer method (RRTMG) for general circulation model (Iacono



et al., 2008) each model hour (two timesteps) with the aerosol optical properties determined from their composition, size, mass, etc.. Specifically, in MAM, the aerosol optical properties (e.g., the specific scattering, specific absorption, and asymmetric parameter) of an internal mixture of aerosol components are expressed in terms of the wet surface mode radius and the wet refractive index. Wet size and volume of aerosol are predicted by assuming the hygroscopic growth following

the κ-Köhler theory (Ghan and Zaveri, 2007) according to the dry radius, density, and hygroscopicity of a particle and the ambient relative humidity and temperature. The wet refractive index is calculated from the composition of the wet aerosol and the refractive index of each component using the volume mixing method. Aerosol optical properties are then parameterized via the Chebyshev polynomial, given the wet surface mode radius and wet refractive index (Ghan and Zaveri, 2007). It is worth noting here that the volume averaging method applied to minerals to compute the bulk aerosol optical

properties may lead to an artificially strong absorption relative to scattering, and thus a low SSA for bulk aerosol (Zhang et al., 2015; Li and Sokolik, 2018). We prescribe the density of each mineral from Scanza et al. (2015) with the exception of goethite, which was not included in that study; the density of goethite is prescribed at 3800 kg m$^{-3}$. The same hygroscopicity (0.068) is assumed for all minerals due to the smaller influence of hygroscopicity on shortwave and longwave radiation compared to other optical properties (e.g., the complex refractive index, dust mineralogy, the size distribution), also

following Scanza et al. (2015). Due to lack of information about the optical properties of chlorite, vermiculite, and mica, we add the mass of chlorite and vermiculite to kaolinite in the clay-sized category, merge chlorite, vermiculite, and mica into one in the silt-sized category, and assume the same optical properties as kaolinite. Such a treatment of these minerals for which the optical properties are missing would not introduce large errors in estimating the dust DRE uncertainty, because 1) they are known to be much less absorbing at the shortwave bands than iron oxides; 2) the dust AOD is insensitive to the

perturbed contents of these minerals within the uncertainty range in soil, since the difference of mass extinction efficiency of these minerals are not that big to make considerable difference on the simulated global dust AOD. Thus, no retuning procedure is required to retain dust AOD of 0.003 in all cases except the ones with high- and low-bounds of dust AOD; 3) our results (Section 3.2.2.1) will also show that the shortwave DRE is insensitive to minerals other than iron oxides, and that the longwave DRE is insensitive to all minerals we considered here. The optical properties of goethite, which is known to

strongly absorb shortwave radiation, differ from those of hematite in terms of both intensity and spectral dependence (Sokolik and Toon, 1999; Lafon et al., 2006). Given no reliable set of spectral optical properties for goethite at bands of our interests, in the base studies using J2014, we assume that goethite is highly absorptive (only second to hematite with the imaginary refractive index of goethite half of hematite), generally consistent to previous calculations (Formenti et al., 2014), and has a hygroscopicity identical to all other minerals.


CAM6 and CAM5(4) are configured with default horizontal resolutions (longitude by latitude: 1.25°×0.9° and 2.5°×1.9°, respectively). All CAM models used 56 vertical layers up to 2 hPa. Meteorology (horizontal wind, air temperature T, and relative humidity) is nudged toward Modern-Era Retrospective analysis for Research and Applications (MERRA) dynamics version2 (CAM6) and version 1 (CAM4 and CAM5), for 2006-2011 with the simulated first year discarded as a model spin-





up period. The nudging is updated with a 6-hour relaxation time scale. We use anthropogenic emissions from AeroCom in CAM4, the Climate Model Intercomparison Program (CMIP5) inventory (Lamarque et al., 2010) in CAM5, and CMIP6 in CAM6, for the year 2000 in all simulations.

The TOA dust DRE under all-sky conditions, unless otherwise stated, is calculated following Eq. (1) as the instantaneous
difference of net fluxes ($\Delta F_{dust}$) at the TOA (Ghan and Zaveri, 2007), diagnosed at each model time step with all aerosol species on the climate diagnostic list ($F_1$) and values with all aerosol species except for dust minerals ($F_2$).

$$\Delta F_{dust} = F_1 - F_2, \qquad (1)$$

### 2.2.2 NASA Goddard Institute for Space Studies ModelE2 (GISS)

The NASA GISS ModelE2.1 has horizontal resolution of 2.5° longitude by 2° latitude with 40 vertical layers extending to 0.1
hPa. In ModelE2.1, dust enters the atmosphere as a result of winds exceeding a prescribed threshold value that increases with soil moisture content. Emitted dust mass is largest within basins where erodible particles have accumulated and there is limited vegetation to protect the soil surface. These regions of preferential emission are identified by Ginoux et al. (2001). Emission depends upon the surface model wind speed and parameterized wind gusts that represent the effects of sub-grid fluctuations (Cakmur et al., 2004). A full model description of emission and transport is given by Miller et al. (2006) with an
updated description of aerosol wet deposition in Perlwitz et al. (2015a).

Prognostic calculation of dust mineral emissions (Perlwitz et al. 2015a,b; Pérez García-Pando et al. 2016) is done based upon the fractional mass abundance of eight minerals within the soil, as derived from measurements of wet-sieved soils by C1999. For particle diameters less than 10 μm, the emitted size distribution of each mineral (except quartz) follows a semi-empirical
fit to measurements (Kok, 2011) that account for the modification of the original soil size distribution by wet sieving. For larger particle diameters (up to 50 μm diameter), the size distribution is constrained from in situ measurements of mineral concentration (Kandler et al. 2009; Pérez García-Pando, personal communication, 2019).

Each mineral is transported separately within five size bins ranging from clay to silt diameters (0.10-2.0, 2.0-4.0, 4.0-8.0,
and 16-32 μm). Goethite and hematite are removed preferentially due to their higher density (about 2-fold) compared to the remaining minerals. Hematite is also transported as a trace constituent as part of an internal mixture with the remaining minerals, allowing hematite to travel farther than in its externally mixed (pure) form. Only mineralogy is predicted in the model, so the DRE is estimated *a posteriori* using the CAM results, as described later.

### 2.2.3 Model Multiscale Online Non-hydrostatic AtmospheRe CHemistry model

The MONARCH model developed at the Barcelona Supercomputing Center (e.g., Pérez et al., 2011; Badia et al., 2017) contains advanced chemistry and aerosol packages, and is coupled online with the Non-hydrostatic Multiscale Model



(NMMB), which allows for running either global or high-resolution (convection-permitting) regional simulations (Janjic et al., 2001; Janjic and Gall, 2012). The dust module of MONARCH (Haustein et al., 2012; Klose et al., in prep.; Pérez et al., 2011) includes different parameterizations of dust emission including those from Marticorena and Bergametti (1995),

Ginoux et al. (2001), Shao (2001, 2004), Shao et al. (2011b), Kok et al. (2014a), and Klose et al. (2014). The model simulations performed for this study utilize the dust emission scheme from Ginoux et al. (2001) with some modifications described in Klose et al. (in prep.). The model includes eight dust size transport bins ranging up to 20 µm in diameter. The emitted size distribution is based on Kok (2011). The inclusion of mineral speciation is under development and therefore it is not included in this study.


The radiation scheme is RRTMG (Iacono et al., 2001, 2008). In the longwave, we assume refractive indices from the Optical Properties of Aerosols and Clouds (OPAC) dataset (Hess et al., 1998) and spherical particle-shape. In the shortwave, we assume tri-axial ellipsoids as described by Kok et al. (2017) who used the dust single-scattering database of Meng et al. (2010). and size-dependent refractive indices based on a globally averaged mineralogical composition. A simple double call

to radiation is used to calculate the DRE for bulk dust. While MONARCH does not calculate mineral speciation of dust, we include its AOD as a measure of uncertainty in comparison to radiative effects related to uncertainty in the soil mineral composition (Fig. 1).

The model is run from 2007 to 2011 at a horizontal resolution of 1.0°×1.4°, with 48 vertical layers. The meteorological fields

are re-initialized daily using ERA Interim reanalysis data (Berrisford et al., 2011), while dust fields and soil moisture are recycled between the daily runs.

## 2.2.4 Geophysical Fluid Dynamics Laboratory model

The latest GFDL global climate model includes the fourth version of the coupled Climate Model (CM4) and Earth System Model (ESM4), with detailed descriptions provided by Held et al. (2019) and Dunne et al. (2019), respectively. In CM4 dust

emission depends only on wind speeds with prescribed dust sources (Ginoux et al., 2001), while in ESM4 it depends also on soil water and ice, snow cover, leaf and stem area indices, and land use type, which are all dynamically calculated, except for land use (Evans et al., 2016). The dust size distribution at emission follows the brittle fragmentation theory of Kok (2011). The simulations were performed from 2010 to 2015 with observed sea-surface-temperature, and sea-ice (i.e. AMIP simulation; Taylor et al., 2000). Dust DRE is not calculated within this model, but the AOD is used to assess the effect cross-

model differences.





### 2.3 Quantifying dust aerosol radiative effect uncertainty

### 2.3.1 Sensitivity studies with mineralogy in the Community Atmospheric Model of version 5

A set of sensitivity studies, based primarily on the CAM5 model, is conducted to characterize the range in DRE due to uncertainties in the soil mineralogical composition. To determine the uncertainty in soil mineralogy, we use two different approaches to estimate the mineral content of soils; the first is based on C1999, and the second based on J2014.

We select simulations with soil mineralogy derived from the MMT of C1999 as the baseline (see Section 3.1 for the resultant hematite aerosol mass percentage). In addition to the mean, the MMT provides uncertainty ranges for each mineral for soil type, for which we calculate the 95% confidence interval of the mineral fraction (Fig. 1). Hematite mass abundance is low but this figure shows that, in general, it has the largest relative uncertainty. Maps containing the upper and lower bounds of minerals, such as hematite, illite, smectite, are similarly created following C1999 using soil type to prescribe mineral fractions. When perturbing the amount of one mineral, we conserve emitted dust mass through an identical and opposite change in soil abundance of the dominant mineral (referred as offsetting mineral) within the same clay- or silt-sized category. Another criterion to select the offsetting mineral is that it should have a minimized impact on the simulated instantaneous TOA fluxes. For example, illite and kaolinite occupy the same clay-sized soil category (0.39) in the calcaric soil type. In this case, we choose kaolinite as the offsetting mineral, because the DRE is less sensitive (measured by the relative change of the DRE over the relative change of the upper-bound kaolinite aerosol content with respect to the base value) to this mineral than to illite in test simulations. Similar to Scanza et al. (2015), a nearest-neighbour algorithm is used to estimate mineral fractions of land mass not specified by the MMT of C1999 in avoid of "zero" dust emissions in these regions. The spatial distribution of uncertainties in the soil mineral abundance based on which we estimate the propagated error in the DRE calculation is discussed in Section 2.3.2.

In addition to C1999, we consider three scenarios based on J2014. One uses the mean mineral fraction from J2014. The other two use low and high bounds on iron oxides. We consider these bounds as the average hematite and goethite mass fractions $\pm 2\sigma$, representing 95% of the variability, where $\sigma$ denotes the standard deviation of hematite and goethite from J2014. The mineral fractions for the rest of the minerals are reduced (increased) proportionally. Compared to clay, there is much less information available for silt-sized minerals and the existing data are obtained mainly based on a number of assumptions rather than observations. Therefore, soil units which do not have an estimate of the uncertainty in the iron oxides, are prescribed to have the maximum uncertainty range that is present in iron oxides across the dataset (Fig. 1). We follow the same procedure as in Section 2.1 to create the global mineralogy atlas. Mineral fractions are normalized to sum to unity.

Table 2 summarizes the experiments undertaken in this study. In the simulations with unperturbed mineralogy (C1999 or J2014), emissions are tuned following Albani et al. (2014) to yield a global mean dust AOD of 0.03 according to the





observational estimate based upon satellite retrievals with bias-corrected AERONET observations and multiple global
models (Ridley et al., 2016). Dust optical properties are based upon Mie Theory which idealizes particles as spheres. In
contrast, AOD retrieved from sun photometers accounts for the dust asphericity (Dubovik et al., 2002). To match modelled
dust extinction with observations, we augment dust AOD globally by ~16% and ~28% for the accumulation plus Aitken and
coarse modes, respectively, according to calculations of Kok et al. (2017), to account for the dust asphericity for the first
time in CAM. We do not consider the increased gravitational setting lifetime due to dust asphericity (Huang et al., 2020),
and leave the lifetime effect of dust asphericity on dust DRE as a future study. Because of the dust AOD augmentation, a
global dust AOD of 0.03 was achieved with a relatively lower dust emission compared to that without considering dust
asphericity. For all other experiments, dust emission is set to be in the same magnitude as in the base except for those used to
assess uncertainty in DRE induced by changing the dust burden.

In order to compare the uncertainty in the DRE from mineralogy to the other effects, we perturb the dust AOD and the
imaginary complex refractive index of the mineral. The dust AOD is perturbed via dust emission adjustment, to be +/-0.005
(in absolute terms), based upon the AOD constraint by Ridley et al. (2016). This perturbation amplitude was also utilized by
Loeb and Su (2010). Considering the variation of dust absorptive properties in different source regions, mainly due to
variations in the iron oxide fraction (Lafon et al., 2006), the imaginary complex refractive index for bulk dust could vary by
up to a factor of two for any given region, while the real part of the index changes less (Kim et al., 2011). Therefore, using a
globally constant imaginary index may not capture a large fraction of the DRE caused by dust minerals. Measurements of the
imaginary complex refractive index also indicate notable differences among different datasets (Zhang et al., 2015a ; Di
Biagio et al., 2019). Here we perturb the imaginary complex refractive index, at the global scale, by +/-16% (relative
percentage) for each mineral, following Kim et al. (2011) whose results are based on AERONET measurements at 14 dust-
dominated sites in and around the Saharan and Arabian Deserts for the sampling period spanning from 1996 to 2009. The
absolute uncertainty (~32%) we considered sits in-between the range of 13-75% for dust aerosol obtained by Di Biagio et al.
(2019).

After undertaking the first set of sensitivity runs, it was found that the calibration of dust AOD inadvertently double counted
the mineral mass, resulting in dust emissions that were too low to obtain an AOD of 0.03 (emission rate of ~3300 Tg a$^{-1}$
compared to ~6600 Tg a$^{-1}$). We reran the model for a second time for those cases (e.g., iron oxides, dust AOD, and
imaginary index) where the perturbed parameter was found to have an important impact on the DRE. The second set of
simulations (dust emission rate: ~4300 Tg a$^{-1}$) introduced the effect of dust asphericity resulting in a global emission increase
of 30% compared to the first set of simulations (dust emission rate: ~3300 Tg a$^{-1}$) with incorrect mass specification for
calculating dust AOD. The comparison of the calculated DRE between the two sets of simulation on the same perturbed
parameter suggests a small difference (global average ≤ 0.05 W m$^{-2}$) (Fig. S1) after applying a "normalization" factor of 1.3.
This factor was determined as the DRE ratio of second to first set of simulations. It approximates the percentage change of





dust emissions between the two sets of simulations (4300 Tg a$^{-1}$/3300 Tg a$^{-1}$), and is comparable with the enhancement of the mass extinction efficiency for particles in the coarse mode to account for dust asphericity. Therefore, we did not repeat those

simulations where varying the minerals did not change the dust DRE. Instead, we use the "normalization" factor to convert the first set of CAM5 simulations (which did not include the shape effect) to the second set (which did include the shape effect). We refer to the simulations that were not repeated in the figures and tables as "normalized" cases.

### 2.3.2 Soil mineralogy uncertainty in C1999

Here we discuss the sensitivity studies with CAM5 using a range of surface mineralogical maps based on the uncertainty in

mineralogical composition by soil type (Fig. 1). Following the methodology described in the previous section and Scanza et al. (2015), multiple soil maps are created and remapped onto CAM5 and CAM6 longitude and latitude grids based on C1999 and J2014 (shown in Fig. S2 for the distribution of minerals in J2014 and in Fig. S3 for the difference between J2014 and C1999) and corresponding soil uncertainties (e.g., Fig. 2). By subtracting the upper-bound mass fraction for each mineral from the base value, we obtain the map of upper-branch uncertainty for minerals such as illite, smectite, hematite and

goethite plus hematite in terms of absolute change (Fig. 2a,b,c,d, also shown is the relative change in e,f,g, and h, respectively).

The amount of soil variability for other minerals tends to be smaller than for iron-oxide and hydroxide elements in terms of relative change (Fig. 2e,f compared to g,h). As shown later (e.g., Section 3.2.2), the iron-oxide and hydroxide minerals are

the most important for the DRE, such that we focus our discussion here on iron-bearing minerals. Our calculation shows that in C1999 hematite, illite, and smectite in clay range between 0.27-0.86%, 9.0-15%, and 6.8-13%, respectively, by mass with a base value of 0.56%, 12%, 10%. In comparison, the globally mean hematite in J2014 is smaller (~0.34%) with an uncertainty range of 0.017-1.0%. Goethite in clay and silt is estimated to be 1.3%, and 0.43%, with a range of 0.36-2.6% and 0.00-1.0%, respectively. We discuss next the spatial distribution of the uncertainty in iron oxides and clays in C1999 and

compare it to that in J2014 (Fig. 2).

#### 2.3.2.1 Iron-oxides

Hematite and goethite are the most common iron oxides present in soils. In-lab analysis shows goethite being less absorptive than hematite (Formenti et al., 2014). Thus, partitioning these iron oxides at emission is relevant to accurately represent the dust DRE in the shortwave spectrum. C1999, however, only considers iron oxides to be in the form of hematite, while J2014

distinguishes two different iron oxide species: hematite (present in the clay size) and goethite (both in clay and silt size fractions) consistent with other measurements (Lafon et al., 2006; Formenti et al., 2008, 2014). Both datasets agree on the scarce mass abundance of iron oxides in the clay- and silt-sized categories as compared to other minerals (note our extension of hematite to the clay-sized category in C1999). The combined iron oxide (hematite and goethite) abundance in J2014 represents a much larger soil fraction than in C1999, particularly in global average (Fig. 2). We found that J2014 shows the





dominance of the iron oxide content by goethite over hematite, regardless of source region. Hematite in C2014 presents strong regional differences as in C1999 with mass fractions predominantly below 1.5%, but in some arid regions, for instance, northern Africa, reaching up to 5.0% (Journet et al., 2014).

C1999 exhibits a large uncertainty in the soil abundance of hematite in the soils of Australia, central and southern Africa,
western India, south eastern part of North America, and eastern Brazil (Fig. 2c). Particularly for areas considered as sand dunes within the Sahel the high-bound hematite in the clay-sized category is ~80% higher (Fig. 2g) than the base. The high iron-oxide content in soils from central Mauritania to central Mali (Lafon et al., 2006; Formenti et al., 2008; Klaver et al., 2011), is represented with a narrow uncertainty range. There is also high confidence in the low iron-oxide fraction attributed to the Bodélé depression (Lafon et al., 2006;  Formenti et al., 2008), which has been characterized by satellite-based sensor
as an active dust source (Ginoux et al., 2012). In J2014, soil abundance of iron oxides is more uncertain than in C1999 over North America, Southern Africa, India, Russia, Western China, and some regions in Europe and Australia. Over most dust source regions, abundance uncertainty in goethite is approximately 1.3 times higher than the base. In contrast, hematite is overall much less uncertain than goethite, and only at some hot spots, it can be 1.6 times higher than the base.

**2.3.2.2 Clays**

Illite dominates the clay size category. Most regions in C1999 show over 25% illite by mass in the clay-sized soils and both atlases report up to 50% clay-sized illite over some Sahara sand dunes. The region-to-region variation for illite is less pronounced than for low-abundance minerals (e.g., feldspars, hematite, and calcite). In comparison to hematite, its soil content uncertainty in terms of the relative change is small (~20%) over dust source areas (Fig. 2e). Large uncertainties primarily exist over regions that tend to have low emissions, such as in southern Africa outside of the Kalahari Desert and
the western part of South America outside of the Atacama Desert (Ginoux et al., 2012). Similarly, smectite abundances are also more certain than hematite, in particular over dust active areas, with a relative change in its soil content less than 10%. Absolute changes of these two minerals, however, are much larger compared to those of hematite in the clay- and silt-sized categories, even in dust source regions. Because of the small influence of these minerals on the shortwave DRE (apparent in C1999 and Section 3.2.2.1), we performed sensitivity tests only on iron oxides but not on illite and smectite when using
J2014.

**2.3.3 Spatially explicit uncertainty estimates**

Spatially, we quantify the contribution of each uncertain parameter described in Section 2.3.1 to the total dust DRE uncertainty by accounting for the deviation in DRE from the perturbed case to the baseline case at target grid boxes. Specifically, the dust DRE due to uncertainties in soil mineralogy (e.g., hematite), are obtained by differencing the DRE
($F_{dust,peturb}$) of each corresponding experiment to that of the baseline simulation ($F_{dust,base}$), as follows





$$\Delta F_{dust,unc} = \Delta F_{dust,peturb} - \Delta F_{dust,base} = (F_3 - F_4) - (F_1 - F_2), \quad (2)$$

where $F_1$ is diagnosed radiative flux at the TOA in the baseline with dust, and, $F_2$ without dust; $F_3$ is diagnosed radiative flux
at the TOA in the perturbed experiment with dust, and, similarly, $F_4$ without dust.

Loeb and Su (2010) had applied the root-mean sum of the squares of the uncertainties associated with each perturbing
experiment (e.g., dust AOD), to get the total DRE uncertainty in global average. This method was also used by Yoshioka et
al. (2006) to estimate the errors for differences between two groups of data. Here, we utilize a similar method and apply it to
the grid cell level to get the total DRE uncertainty (Eq. (3) for C1999 and Eq. (4) and Eq. (5) to account for difference
between the two soil datasets) due to parameters we considered (minerals, dust burden, or imaginary complex refractive
index for each mineral). Our adopted method, firstly, indicates an assumption that any difference between the experiment
and base on the DRE calculation belongs to a part of the overall uncertainty and thus should be accounted for at the grid cell
level (Eq. (3), and Eq. (4) and Eq. (5)), and, secondly, effectively assumes the perturbed parameters are independent. As in
Loeb and Su (2010), we separate cases with a stronger warming from those with the opposite effect, splitting uncertainty into
low and high branches, but at the grid cell level. These branches show the maximum range of DRE that we can achieve
through any combination of our perturbed experiments, assuming that these perturbations are independent.

$$\begin{cases} \Delta F_{hig} = \sqrt[2]{\sum_{i=1}^{n-3}(\Delta F_i - \Delta F_{base})^2} \quad, \quad \Delta F_i \geq \Delta F_{base} \\[4mm] \Delta F_{low} = \sqrt[2]{\sum_{i=1}^{n-3}(\Delta F_i - \Delta F_{base})^2}, \quad \Delta F_i < \Delta F_{base} \end{cases}, \quad (3)$$


$$\begin{cases} \Delta F_{hig} = \sqrt[2]{\sum_{i=1}^{n-3}\left(\Delta F_{C,i} - \Delta F_{C,base}\right)^2 + b_{hig}^2} \quad, \quad \Delta F_i \geq \Delta F_{base} \\[4mm] \Delta F_{low} = \sqrt[2]{\sum_{i=1}^{n-3}\left(\Delta F_{C,i} - \Delta F_{C,base}\right)^2 + b_{low}^2} \quad, \quad \Delta F_i < \Delta F_{base} \end{cases}, \quad (4)$$

where $\Delta F_{upp}$ and $\Delta F_{low}$ represents uncertainty in absolute terms; subscript "hig" and "low" show high and low branches; n is
the total case number; i indicates different cases; "base" refers to the baseline simulation (CAM5 with C1999); "C" denotes


C1999; n-3 means that we exclude three cases associated with J2014 (see Section 2.3.1); term "b" is calculated following Eq. (5).

$$
\begin{cases}
{b_{hig}}^2 = \left(\Delta F_{J,base} - \Delta F_{C,base}\right)^2 + \sum_{i=1}^{2}\left(\Delta F_{J,i} - \Delta F_{J,base}\right)^2, \ \Delta F_{J,base} \geq \Delta F_{C,base} \\
{b_{low}}^2 = \left(\Delta F_{J,base} - \Delta F_{C,base}\right)^2 + \sum_{i=1}^{2}\left(\Delta F_{J,i} - \Delta F_{J,base}\right)^2, \ \Delta F_{J,base} < \Delta F_{C,base}
\end{cases} , \quad (5)
$$

where "J" represents cases with J2014, and the "2" cases are for oxides with maximum and minimum soil abundances. Eq (3) includes only the perturbations to the model based upon C1999. Eqs. (4) and (5) allow the inclusion of perturbations associated with the J2014 soil mineral content. The $b_{hig}$ and $b_{low}$ factors allow the effect of perturbations calculated using J2014 to be included in the total DRE uncertainty despite the different base state of this model compared to that calculated using C1999.


We do not quantify the global mean uncertainty by simply averaging the value we obtained at all grid boxes, because there is no simple relationship between local and global uncertainty. Local uncertainty correlates across neighbouring grid boxes, and this correlation probably varies spatially. Therefore, a simple average of the local deviation would very likely lead to bias in the global mean estimate toward regions with large correlation. Instead, we characterize global average uncertainty of

the DRE based on the global mean of different cases as in Loeb and Su (2010).

In addition to the total DRE uncertainty due to all parameters considered, to quantify the contribution of uncertainty in the soil distribution of iron oxides to the total uncertainty, we repeat the above calculation but single out the effect of iron oxides.

**2.3.4 Estimating radiative effect from other models**

In order to understand the relative importance of uncertainties in mineral amounts to other uncertainties in dust DRE, we require estimates of the DRE from other model estimates, using up-to-date dust optics and size distributions, but there are limited models available that simulate mineral distributions. At present, the relation of dust mineral composition to AOD and DRE in ModelE2 is under development. Instead, we predict the shortwave dust DRE (under all-sky conditions unless

otherwise stated) assuming that the relationship between the DRE and the monthly column hematite mass in CAM5 also holds in ModelE2. This relationship is founded by applying a least squares regression to each grid cell based on the monthly DRE and column hematite mass in a CAM5 case with the high-bound hematite in the clay-sized category. We select the CAM5 high-bound case, because it simulates a similar global hematite loading as in ModelE2. The regression model only



includes hematite because the shortwave DRE is most sensitive to it. This is supported by various laboratory experiments of
dust samples (Moosmüller et al., 2012; Di Biagio et al., 2019), and will be discussed further in Section 3.2.2.1.

As a test of the regression model, the DRE derived solely from hematite mass in CAM5 shows good agreement and self-
consistency with the actual DRE (Fig. S4a,b). The predicted DRE aligns well with the actual value: the global mean
difference is +0.01 W m$^{-2}$, a measure of the uncertainty of our estimates of the DRE based upon the GISS ModelE2. The
regression process reproduces the spatial contrast of aerosol warming and cooling. When applying the slope to CAM6 (Fig.
S5a,b), the biases are larger along the "dust belt" (Fig. S5a) with positive errors over regions such as the Sahel, and negative
errors across most of the Sahara Desert (Fig. S6a).

Similarly, the shortwave DRE in GFDL is predicted based on its simulated bulk dust AOD (i.e., without mineral speciation)
using the least squares regression derived from CAM5. To make the models more comparable, we increase the dust amounts
in the GFDL model by 1.5, so that the AODs are both ~0.03 (Table 3). We compute the regression slope and interception
based on the shortwave DRE and dust AOD in the CAM5 baseline. This approach works well for CAM5 (Fig. S4c,d) and
CAM6 (Fig. S5c,d.), and yields a similar DRE between GFDL and CAM5 with the global mean difference <0.08 W m$^{-2}$
(Fig. S6b). To check how the approach works for non-CAM models, we show the comparison for MONARCH, where we
know the DRE (Fig. S7). In this case, there are some differences spatially, as the regression model underestimates the dust
cooling over North Africa, the Middle East and Central Asia. Globally, the underestimation reaches up to ~0.2 W m$^{-2}$.

## 3 Results and Discussion

### 3.1 Simulated atmospheric mineral concentration uncertainty

Once dust is emitted, the uncertainties in the soil mineral abundance (see Section 2.3.2) propagate into the uncertainties in
the simulated atmospheric dust aerosol mineralogical composition. Table 4 lists the base global mean atmospheric dust mass
fractions for hematite (1.7%), illite (27%), and smectite (18%), and their uncertainty ranges of 1.1-2.2%, 22-32%, and 13-
23% (absolute changes of lower- and upper- bounds with respect to the base), respectively, in CAM5 using C1999. The
uncertainty range in hematite in the clay soil fraction (0.27-0.86%) results in approximately a 35% relative change in its
simulated atmospheric burden with respect to the base; this value is 18% for illite, and 26% for smectite (Table 4). The
brittle fragmentation theory used on the fully disaggregated soil particles puts clay-sized soil particles 130% more into
coarse-mode aerosol particles (Table 2b of Scanza et al., 2015) increasing the baseline percentage of silt-sized aerosol
hematite. Consequently, uncertainty in hematite in the clay-sized soil category leads to a larger relative change in simulated
total hematite burden compared to silt (35% versus 13%, respectively), although identical soil uncertainties are prescribed.
Similar results are obtained in CAM4, because it is binned with the same diameter bounds as in CAM5 (bin 1: 0.10-1.0 µm
in CAM4 versus Aitken and accumulation modes in CAM5, and bin 2-4: 1.0-10 µm versus the coarse mode). CAM6





simulates a much smaller hematite fraction of the total dust mass as we prescribed hematite solely from the clay-sized soil, despite similar values for illite and smectite fractions. Silt-sized soil hematite sources were removed for a CAM6 sensitivity test, because its omission improved the model-observation comparison in SSA for CAM5 (Scanza et al., 2015), and also in the clear-sky DRE efficiency, which will be discussed in Section 3.2.2.2. Combining all three versions of CAM yields an

estimate of the global mean hematite burden of 0.58-2.2% of the total dust by mass.

Perturbing hematite in the silt- and clay-sized categories requires an opposite and compensating change in the abundance of the remaining minerals in the same soil-sized category (Section 2.3.1), which are often dominated by phyllosilicates (e.g., illite, kaolinite, and smectite) (Claquin et al., 1999). As iron oxides are, in general, a small fraction of dust mass, this change

represents a tiny fraction for the offsetting mineral, generally less than ~2% in practice. Table 4 and Fig. 2a,b,c show that absolute uncertainty in the hematite change in C1999 is frequently much smaller than that of either illite or smectite with the absolute change of simulated hematite aerosol mass fraction with respect to the base value (1.7%) is ~0.6% and ~0.3% due to uncertainty in the clay- and silt-sized category, respectively. The simulated relative change for hematite is comparable to kaolinite, large compared to illite, smectite, quartz, and feldspar, but small compared to gypsum.


We show spatial distributions of the relative change of simulated mass fraction due to uncertainty in iron oxides in both two atlases and kaolinite in C1999 in Fig. 3 (other minerals, see Fig. S8), and the column mean mineral mass percentage simulated in CAM5 and CAM6 in Fig. S9. North Africa, in particular the Sahel, followed by the Middle East, are important dust sources of hematite (Fig. S2d,h,n). In agreement with the location of the maximum hematite fraction observed in soils

within C1999, large mean column hematite fractions are found in the interior of Australia and to its north (Fig. S9k), and in the dust plume that extends from North Africa to South America. The high hematite content in dust particles from the Middle East agrees with Kruger et al. (2004). The comparison of iron oxides with other minerals in global average (e.g., the smaller absolute uncertainty in hematite change comparable to other minerals and comparable relative change between hematite and kaolinite) is somewhat true regionally (Fig. S8). For example, over North Africa and the dust plume in

downwind regions, uncertainty in the soil abundance of hematite in the clay-sized category in C1999 leads to a relative change of ~40% in the atmospheric abundance compared to the baseline simulation in CAM5. This regional relative change of simulated hematite aerosol mass fraction is a little small compared to kaolinite (Fig. 3), large compared to illite (Fig. S8e), smectite (Fig. S8g), and quartz (Fig. S8i,j).

In addition to the variation in soil mineral distribution, the uncertainty in the monthly mean mineral composition of dust aerosol is sensitive to the seasonal cycle and the interannual variability in dust emissions (Smith et al., 2017) as well as the model version used. Fig. S10c,d shows the coefficient of variation (CV: calculated as the ratio of the standard deviation of the monthly means to the mean across all experiments, including results from GISS ModelE2) for iron oxides. The global mean CV is less than 1.0. In the regions that are downwind of the major dust sources, except the Patagonian Desert and


Australian deserts, variability in the iron oxide(s) amount (CV<0.9) is lower than that which occurs over the Sahel Desert and dust sources in Australia, likely due to seasonal and interannual variability of the dust emissions (e.g., Mahowald et al., 2003). Much of the variability shown in Fig. S10 is due to including results from different models, as seen by contrasting the CV of the combined CAM5, CAM6, and ModelE2 (e.g., Fig. S10c; global mean CV equals 0.7) to those of CAM5 only (Fig S11) (e.g., Fig. S11c; global mean CV equals 0.3). The effect of model differences on the hematite variability is also

illustrated in Fig. S12. The combined CV between different models (e.g., Fig. S12a, global mean CV equals 0.6 for combined CAM5 and CAM6; Fig. S12b, 0.7 for combined CAM5 and ModelE2) is larger than that induced by the soil uncertainty in hematite in C1999 in CAM5 only (Fig. S11c), but comparable to the CV (Fig. S11d; global mean CV of 0.7) obtained accounting for the difference between the two atlases. Despite matching soil mineralogy C1999, ModelE2 and CAM5 differs in various aspects of dust mineral representation including the treatment of aerosol mixing states for mineral

species. Specifically, ModelE2 represents hematite in both the pure crystalline form (externally-mixed, as for CAM4) and as small impurities attached as an internal mixture to non-iron oxide minerals (internal-mixed, as for CAM5/6). Hematite aerosol in the pure form is removed quickly from the atmosphere by gravitational setting because of larger density (5260 kg m$^{-3}$) compared to other minerals (density <4000 kg m$^{-3}$). In contrast, the allocation of hematite within a mixed aerosol composition facilitates long-range transport of the hematite contained within, because hematite occupies only a small mass

(volume) fraction and thus the aggregated density is determined by the host mineral (s). Due in part to the different treatments of hematite between CAM5 and ModelE2, combined variability between CAM5 and ModelE2 (global mean CV=0.7) is comparable to that due to uncertainty of iron oxides in the two atlases (global mean CV=0.6) and also comparable to a combination of CAM5 and CAM6 with removed hematite source from the silt-sized category (global mean CV=0.6).

**3.2 Shortwave direct radiative effect uncertainty**

**3.2.1 Base simulation direct radiative effect**

The choice of the soil minerology dataset and model employed has a strong impact on the derived DRE (Table 5 and Fig. 4). CAM5 with C1999 simulates a global mean TOA DRE of -0.18 W m$^{-2}$ compared to -0.34 W m$^{-2}$ in CAM6 (Table 5 and Fig. 4b). Compared to the CAM5 baseline, CAM4 has a similar global mean TOA DRE (-0.13 W m$^{-2}$) assuming external mixing

compared to the internal mixtures of CAM5. However, CAM4 simulates a different spatial pattern with more warming over the majority of the North Africa deserts consistent with Scanza et al. (2015). They obtained a slightly less global shortwave cooling compared to CAM4 results of this study, because of lower dust AOD (0.016) only half the value in this study (0.03). Note that in CAM4 the optical properties for minerals (quartz, gypsum, feldspar, and calcite) are calculated considering an internal mixture in both Scanza et al. (2015) and this study. In contrast to the similarity of DRE between CAM4 and CAM5,

CAM6 with C1999 simulates a stronger global averaged shortwave cooling of -0.34 W m$^{-2}$ and more areas showing cooling effects (Fig. 4b,c), because we assumed hematite solely comes from the clay-sized soil category, resulting in a smaller



hematite aerosol mass fraction (CAM5: 1.65%; CAM6: 1.11%). The treatment of iron oxides within the model is therefore important for estimates of the dust DRE.

Regionally, the mean shortwave dust DRE for the base simulation shows warming over North Africa and cooling downwind (Fig. 4a,b), similar to previous studies (Miller and Tegen, 1998; Yoshioka et al., 2006) and other model versions used in this study (CAM6 in Fig. 4b). We find that in the baseline where the annual mean surface albedo exceeds 0.2 in the visible spectrum shortwave DRE is positive, and negative otherwise. There is also a strong warming contribution over desert land regions such as North Africa and the Middle East compared to remote regions due to a higher shortwave absorbing

efficiency of large-sized particles (Kok et al., 2017) which are found at a relatively larger fraction close to the emission source (Mahowald et al., 2014; Ryder et al., 2019). These simulations underestimated coarse dust (diameter between 5-10 μm) and missed the very coarse dust (diameter >10 μm), as well underestimated transport of particles >5 μm in diameter further away from the source (Ryder et al., 2019; Adebiyi and Kok, 2020). This underestimation of the coarse and very coarse dust particle transport may result from inaccurately representing turbulent or convective vertical mixing that could

decrease the dry deposition of dust aerosols (Adebiyi and Kok, 2020) and from not accounting for the dust asphericity which can increase the gravitational settling lifetime (Huang et al., 2020). Consequently, we may underestimate both the shortwave and longwave (Section 3.3) dust warming.

Comparing the shortwave DRE from CAM5 simulations with different mineral maps, C1999 and J2014 (Fig. 4d), shows a

slight difference in the DRE amplitude at the global annual mean scale (-0.18 W m$^{-2}$ versus -0.14 W m$^{-2}$, Table 5). However, there are noticeable regional differences comparable in amplitude to the DRE itself. J2014 contains larger soil fractions of iron oxides (sum of hematite and goethite) within main dust source regions like North Africa (Fig. S3d,e,l). A more positive DRE is thus realised when using J2014 compared to C1999 over most dust-dominant continents and even oceanic regions such as the North Atlantic Ocean.


Previous studies (Sokolik and Toon, 1999; Lafon et al., 2006) have shown that hematite and goethite have distinct optical properties at the shortwave bands. Considering both hematite and goethite in mineral dust produced a more flat spectral SSA, owing to the less pronounced dependence of the imaginary refractive index of goethite on the short wavelengths (Formenti et al., 2014). If we assume that goethite is less absorbing than hematite, we obtain a global mean shortwave DRE of -0.14 W m$^{-2}$

(Table 5). Assuming goethite is as absorbing as hematite leads to an even larger increase of the shortwave DRE: -0.05 W m$^{-2}$ (Table 5; "Same hem and goe"). The 64% reduction in the shortwave cooling is thus due to the stronger absorption of shortwave radiation by hematite than by goethite. Over the North African continent, distinguishing the optical properties of these two iron oxides produces a difference of ~56% in the shortwave DRE.





### 3.2.2 Uncertainty of shortwave direct radiative effect and importance of iron oxides

In this section, we characterize the shortwave DRE uncertainty due to dust minerals, dust burdens, imaginary refractive index of minerals, and radiative parameterization, while other uncertainty sources are discussed in Appendix A. We evaluate the importance of iron oxides upon the shortwave DRE variation relative to other minerals, dust burden, and the surface albedo. Shortwave DREs from multiple models are compared and included in the shortwave DRE estimate based on the methodology described in Section 2.3.3. Scanza et al. (2015) showed a model-observation comparison of the clear-sky

shortwave DRE efficiency calculated with earlier versions of mineralogy CAM4 and CAM5, as well as released versions of both models. With updated mineralogy in CAM5, as well as ported mineralogy in CAM6, we revisit the model-observation comparison in this section by also including the uncertainty in iron oxides derived from the soil abundance in C1999 and J2014.

### 3.2.2.1 Uncertainty due to dust minerals, burden, and imaginary complex refractive index

The sensitivity studies undertaken with CAM5 (Table 2) show that the uncertainty of hematite causes the largest change in the global mean dust DRE (Table 5 and Fig. 5a) and SSA at the 0.44-0.63 µm band (Fig. 5b) compared to the uncertainty of other minerals. Scanza et al. (2015) showed that in CAM5 with hematite confined solely from the clay-sized category, the sign of the dust DRE at TOA is altered from slightly positive (+0.05 W m$^{-2}$) in CAM5 with hematite confined from both clay and silt categories to slightly negative (-0.04 W m$^{-2}$), despite similar surface DRE (not shown), suggesting the importance of

hematite in the shortwave DRE estimate at the TOA. Fig. 5a and Fig. 6 demonstrate the importance of hematite for the TOA DRE. We see a more (less) cooling value of -0.28 W m$^{-2}$ (-0.10 W m$^{-2}$) in CAM5 with the low (high) bound of hematite in the silt-sized category compared to the baseline simulation (-0.18 W m$^{-2}$) resulting from the changed SSA (Fig. 5b). Similarly, use of the high bound of clay-sized hematite significantly decreases SSA (Fig. 5b), leading to even less cooling (-0.08 W m$^{-2}$), compared to lowering the clay-sized abundance. We can thus expect that the larger uncertainty in iron oxides in

J2014, compared to that in C1999, would lead to a larger range in the global annual mean SSA and thus a larger uncertainty in shortwave DRE. The importance of hematite for shortwave DRE is true regionally as well as globally (Fig. 6). Uncertainty in other minerals in the soil causes a small change of the shortwave DRE globally (less than 10% of the uncertainty related to hematite; Fig. 5a) due partly to their small fractional change relative to the large total abundance of those minerals in terms of the soil distribution (Fig. 1) and to their low shortwave DRE efficiency (Fig. 7a). For example, increasing the soil amount

of illite to its high bound results in an additional warming of +0.01 W m$^{-2}$ (for other minerals see Fig. 5a). Fig. 6a and Fig. S13a,b (other minerals, see other panels of Fig. S13) show that an increase of hematite in either the clay-sized or silt-sized soil categories leads to more warming over both continental and downwind oceanic regions at the TOA and vice versa, which is consistent with the absorbing nature of iron oxides and with results of previous sensitivity studies (Balkanski et al., 2007). This influence of hematite aerosol burden on the shortwave DRE is most apparent over North Africa, in particular





over the western Sahara and Sahel (e.g., Fig. 6a), where a large uncertainty exists in the underlying hematite soil abundance in C1999 (Fig. 2c).

The response of shortwave DRE to increasing dust AOD to the upper limit (0.03+0.005; Ridley et al., 2016) has a very different spatial pattern (Fig. 6c) in comparison to perturbing hematite abundance or the imaginary refractive index of 665    minerals (Fig. 6a,d). For example, increasing iron oxide content results in a uniformly stronger warming, owing to the enhanced ability of dust aerosol to absorb shortwave radiation. A higher dust AOD, however, tends to enhance the warming-to-cooling contrast, given a certain emission scheme, by amplifying the baseline shortwave DRE (Fig. 4a) due to more total surface area to absorb and/or scatter shortwave radiation, whose features depend on the annually mean surface albedo.

Fig. 7 displays the sensitivity of the DRE at the TOA to dust AOD, imaginary indices, and the mineral content in soil in CAM5 with C1999. The sensitivity in Fig. 7 is calculated as the ratio of the relative change of the DRE to the relative change of each driver, both with respect to base simulation values. The DRE is most sensitive to changes in hematite in the silt-sized category. In contrast, perturbations to other minerals, including illite and smectite, within their 95% intervals, have a small influence on the shortwave DRE in terms of the globally averaged value owing to negligible resultant changes in the SSA 675    (Fig. 5b). The cancelling of opposite regional effects (Fig. 6c) by perturbing dust AOD over regions with low (annual mean ≤0.2; negative DRE) and high (annual mean >0.2; positive DRE) visible surface albedo results in little change of the global mean shortwave DRE (Fig. 5a), although regional changes and especially land-sea contrasts may be larger. Consequently, a large fraction of total uncertainty in the global mean DRE is attributed to uncertainty in the soil hematite because of its higher absorption efficiency at the shortwave bands.


The estimated uncertainty due to all effects combined is illustrated in Fig. 8 based on the method described in Eqs. (3) through (5) of Section 2.3.3. For low-bound uncertainty, we only show in Fig. 8 the global mean value (inlet numbers), because of a reginal similarity of the two bounds. Globally, we obtain a total range of [-0.12, +0.11] W m$^{-2}$ based on uncertainty of mineral distribution in C1999, dust AOD, and imaginary indices. Perturbations on iron oxides in the clay- and 685    silt-sized categories result in an uncertainty range of [-0.11, +0.09] W m$^{-2}$, contributing ~87% of the total range. Adding the difference between the mineral distribution in C1999 and in J2014, and the iron oxide uncertainty in J2014 yields a larger total uncertainty range of [-0.23, +0.28] W m$^{-2}$. The majority of the total uncertainty including both the C1999 and J2014 experiments (~96%) can be attributed to uncertainty in soil fractions of iron oxides, considering the resulting range of [-0.22, +0.27] W m$^{-2}$ due to iron oxides only. We find that the spatial pattern of this high-bound uncertainty in C1999 is similar to 690    that of the intensified warming due to solely more hematite in Fig. 6a. Because a similar spatial distribution presents in both low and high uncertainty bounds, large absolute uncertainties occur over North Africa, specifically over regions spanning from Mauritania through Niger and Chad to Sudan.



In CAM4, which employs an external aerosol mixing assumption, there is a lack of sensitivity in the shortwave DRE to any
mineral (Fig. S14). Perturbating hematite produces a small change of SSA within 1% (relative change, not shown) and hence
a small change of the shortwave DRE (Fig. S14). Because of this, previous results using CAM4 were also insensitive to
changes in hematite aerosol burden (Scanza et al., 2015). Results from this study are consistent with Sokolik and Toon
(1999), who demonstrated that to have SSA lower than 0.9 at 0.50 µm requires an unrealistically high amount of hematite
under the external mixing assumption. Reduced DRE sensitivity to variations of the hematite by external mixtures of
hematite (compared to internal mixtures) has also been shown by Koven and Fung (2006) and Balkanski et al., (2007).

### 3.2.2.2 Clear-sky shortwave radiative effect efficiency: model to observation comparison

There are limited calculations of the DRE efficiency estimated from satellite retrievals that can be used for comparison with
model results. Fig. 9 compares the TOA DRE efficiency of dust under clear-sky conditions (W m$^{-2}$ τ$^{-1}$; defined as the ratio
of clear-sky DRE to dust AOD) obtained with mineralogy in CAM5 and CAM6 to clear-sky satellite-based observations
over the Sahara. Over the ocean, both models and most cases yield a DRE efficiency which is not significantly different from
observations during summer and winter (Li et al., 2004). According to Patadia et al. (2009), the observed clear-sky DRE
efficiency over North Africa is approximately zero for a surface albedo of 0.4 during the "high" dust season (June, July and
August: JJA). Compared to observations, both models with C1999 yield a similar shortwave clear-sky DRE efficiency.
CAM5 with the high-bound iron oxides, as derived by the uncertainty in J2014, shows larger values, possibly suggesting too
much shortwave absorption. Of the two models and different cases considered here, CAM6 with C1999 has more skill in
better reproducing observations of the DRE efficiency. All models underestimate the longwave DRE efficiency in September
when compared to the observation (Zhang and Christopher, 2003), although we augmented the longwave DRE by 51% to
account for dust scattering neglected by CAM (Scanza et al., 2015).

### 3.2.2.3 Understanding the relative roles of single scattering albedo, hematite, and dust AOD on the shortwave direct
radiative effect

A fundamental question for this study is: what are the most important determinants in altering the shortwave DRE for
different regions? Analysis of soil samples taken from locations representative of the Sahara and Sahel deserts suggest that a
linear correlation exits between SSA and the iron content in fine sized dust particles (<2 µm in diameter) at visible and
infrared bands (Moosmüller et al., 2012). A recent study built on this showed that the relationship is statistically significant
at all shortwave wavelengths and not limited to fine sized dust (Di Biagio et al., 2020). The relative shortwave absorption
(related to SSA) of dust particles should thus be related to iron oxide burden, in addition to its dependence on dust size
distribution and effects upon the complex refractive index by other minerals. Here, we use variations across different
experiments, and interannual variability in our model simulations to assess the relative roles of iron oxides (Fig. 10), dust
AOD, and surface albedo (Fig. 11) over different regions.




First, we consider the relationship of derived dust SSA at the 0.44-0.63 μm band to the hematite mass fraction over dust-dominated areas, both globally and over five sub-continental regions containing major dust sources (North Africa, Middle East, Central Asia, North East Asia, and Australia; domains defined in Table S1 and dust sources shown in Fig. 10). As the SSA calculated in CESM is for all aerosols, we extract the dust SSA following Scanza et al. (2015) by only selecting those
grid boxes where the ratio of dust AOD to total AOD is greater than 0.5 ($AOD_{dust} > AOD_{total} \cdot 0.5$; the derived SSA varies only a little bit with a higher fractional threshold of 0.8; Table 5) and the land coverage is 100% of the total grid box area.

Fig. 10 illustrates a strong regional variability of the derived dust SSA at the 0.44-0.63 μm band, hematite mass fractions, and their relationship to one another. The quantitative analysis shows a statistically significant negative relationship between
global mean SSA and the hematite mass fraction for both coarse (Pearson correlation: R=-0.92) and fine (sum of Aitken and accumulation, R=-0.87) modes over land grid pixels at the 95% confidence level (student's t-test). Dust SSA is more closely correlated with the coarse-mode hematite mass fraction over North Africa and Australia, and more closely correlated with the fine-mode hematite mass fraction for the Middle East, Central Asia, and North East Asia. The modelled SSA over dust dominant areas ranging between 0.83-0.91 (Table 5) revealed high absorption by dust at this band. Three aspects may
explain the low SSA. Firstly, the criterion for removing non-dust aerosols, which excludes pixels with $AOD_{dust} \leq AOD_{total} \cdot 0.5$ passes absorptive non-dust aerosols. Secondly, the use of the volume averaging of minerals to compute the complex refractive index for bulk dust could yield an artificially strong absorption compared to scattering and thus low SSA (Zhang et al., 2015; Li and Sokolik, 2018). In contrast to these two aspects, the underestimation of coarse dust particles (>5 μm) could bias SSA toward high values, because of the large surface area of coarse dust particles for radiation absorption. All the three
aspects could influence the accuracy of the derived dust SSA and thus its relationship with hematite aerosol. Nevertheless, our results regarding the relationship between SSA and hematite mass fraction agree with Moosmüller et al. (2012) and Di Biagio et al. (2020). The coexisting of dust and absorptive non-dust aerosol (e.g., black carbon; Kim et al., 2004; Ge et al., 2010) could partially explain the "discrepancy" between the low derived dust SSA and the relatively strong shortwave cooling by dust over North East Asia (Fig. 11; the shortwave DRE versus dust AOD) despite the low derived dust SSA. The
correlation between SSA and hematite mass fraction statistically highlights the importance of the simulated hematite for the shortwave dust DRE estimate. It suggests that over most dust source regions the shortwave DRE uncertainty due to iron oxides in C1999 and J2014 significantly (p-value<0.05, student's t-test) exceeds the annual mean shortwave DRE by 2σ (Fig. S15), where σ denotes the standard deviation of the annual mean DRE with the seasonal cycle removed.

Fig. 11 shows response of the variability of shortwave DRE to that of dust AOD and the surface albedo globally and over the examined sub-regions. Over all sub regions except Australia, the variability of shortwave DRE is statistically significantly (p-value<0.05, student's t-test) correlated with that of dust AOD. The relationship between these two variables is regionally specific with different slopes for difference regions (Fig. 11a), mainly depending on the annual mean surface albedo (Fig. 11b). For regions such as North Africa and the Middle East with an annual mean surface albedo of 0.28 in CAM5, shortwave





DRE positively scales with dust AOD, because the shortwave DRE is dominated by dust absorption over surfaces with the annual visible surface albedo >0.2. In contrast, the shortwave DRE inversely scales with dust AOD in Central Asia and North East Asia, where the annual visible surface albedo <0.2 and the shortwave dust scattering dominates over absorption. This is the same as the influence of dust AOD on the shortwave DRE from a climatology perspective: intensified warming (cooling) over a region where the shortwave DRE is positive (negative) in the baseline simulation (Fig. 6c). The surface

albedo variability in North Africa and Middle East is weak compared to other sub regions. Overall, dust DRE becomes more warming (less cooling) as the surface albedo increases due to the absorption of more reflected shortwave radiation, consistent with the results of previous studies ( Liao and Seinfeld, 1998; Miller et al., 2014; Li and Sokolik, 2018).

### 3.2.2.4 Model diversity: across model comparisons

Previous studies have highlighted how the variability in the DRE is due to different model representation of the sensitivity of

DRE to dust minerals, dust optical properties, surface albedo, and aerosol-cloud interactions (Huneeus et al., 2011; Shindell et al., 2013; Kok et al., 2017). We estimate in this section the multi-model spread in the shortwave DRE using both soil mineral distributions based on all experiments (Table 2) at each grid cell.

The shortwave DRE from ModelE2 is not directly calculated based on the model run but derived here *a posteriori* via

regression (see Section 2.3.4). Globally, the predicted shortwave DRE (-0.09 W m$^{-2}$) is less negative than in the CAM5 baseline (-0.18 W m$^{-2}$). We derive a stronger warming over most desert areas in ModelE2 than in CAM5 with C1999 (Fig. S16a). The strong warming in ModelE2 compared to CAM5 highly likely results from the high hematite aerosol mass simulated in ModelE2 over the Sahel desert, the Middle East, and Australia (Fig. S17), although the regression model induced error may also contribute (Fig. S7a). Similarly, we use the dust AOD distribution in the GFDL model to estimate the

shortwave DRE (described in Section 2.3.4). The resultant estimate (-0.23 W m$^{-2}$) is slightly lower than that in our base case in global average (Fig. S6b). Over most desert regions in North Africa, the Middle East, and Central Asia, GFDL dust shows stronger cooling compared to the CAM5 baseline.

Dust DRE from MONARCH is calculated by the model and reported here. In global average, MONARCH simulates a

stronger cooling (-0.37 W m$^{-2}$) compared to CAM5 with C1999 (-0.18 W m$^{-2}$; Fig. S16b) partly due to a more scattering dust in the former (SSA: 0.92 and 0.89 in MONARCH and CAM5, respectively; Table 3). The stronger cooling is seen most clearly over the land areas in North African and the Middle East (Fig. S16b).

We estimate the DRE uncertainty to be [-0.23, +0.14] W m$^{-2}$, considering the combined model spread (CAM5, CAM6,

ModelE2, MONARCH, and GFDL) and uncertainties in the soil mineral abundance in C1999, dust burdens, and imaginary refractive index of minerals. This range is even narrower than the uncertainty induced by all parameters that we have considered in the perturbation analysis using CAM5, implying that the effect of inter-model differences is smaller than the





uncertainty revealed by CAM5, even though the *a posteriori* statistical DRE calculation for ModelE2 and GFDL models introduces uncertainties. Adding the difference between C1999 and J2014, and iron oxide uncertainty in J2014 to the result

broadens the range to be [-0.30, +0.30] W m$^{-2}$. Therefore, even considering the model spread, iron oxides are still the most important error source in terms of the contribution (82%=(0.22+0.27)/(0.30+0.30)×100; cf. Section 3.2.2.1 for the numerator) to the total shortwave DRE uncertainty. Spatially, the total shortwave DRE uncertainty (Fig. 12) including the model spread is in general larger than that due to soil iron-oxide uncertainty in C1999 and in both two datasets, particularly over the Middle East, western North Africa, and oceanic areas downwind of North Africa.

**3.2.2.5 Errors in shortwave direct radiative effect calculations due to radiative parameterization**

The band error in the model radiation parameterization in the model is an important uncertainty source for the DRE estimate (Jones et al., 2017). We assess this uncertainty with a line-by-line calculations using the CAM model (e.g., Jones et al., 2017) for a one-day (March 22nd, 2005) simulation over North Africa. According to the line-by-line calculation, the shortwave bands implemented into CESM introduce negative bias (~25% error) in the TOA DRE calculation compared to

the benchmark radiation code (a similar error level is shown in the TOA DRE calculation under clear-sky and all-sky conditions; Paynter, personal communication, 2020). This suggests that despite the use of accurate optical properties, these GCMs underestimate the DRE and dust warming mostly due to the use of the two-stream δ-Eddington approximation in RRTMG and the radiative model's low band resolution (Paynter, personal communication, 2020). The underestimation, however, is small with an amplitude of ~0.05 W m$^{-2}$ considering the DRE in our baseline simulation. Thus, although the

line-by-line calculation is performed only for one full day over North Africa, we suggest that the uncertainty associated with the band error in GCMs is likely much smaller than that due to iron oxides (Section 3.2.2.1).

**3.3 Longwave radiative effect uncertainty**

CAM5 simulated differences in the longwave dust DRE. Unlike the shortwave DRE, the longwave DRE uncertainties mainly arise from the uncertainties in the mineral complex refractive indices, size distribution, and vertical distribution

(effectively, dust acts similarly to a greenhouse gas) of dust aerosol rather than mineralogy. Our sensitivity tests show that the longwave DRE is insensitive to the change of dust mineral contents either in the clay- or silt-sized category (Fig.13). The global mean longwave DREs calculated by different CAM versions are +0.24 W m$^{-2}$ in CAM4, +0.11 W m$^{-2}$ in CAM5, and +0.14 W m$^{-2}$ in CAM6.

Our calculation suggests negligible impacts on the longwave DRE by uncertainty in the soil distribution of minerals such as quartz and feldspar (Fig. S18), which may be a result of the longwave bands and the averaged absorption properties of the eight minerals used in CAM5. Quartz dominates absorption at several longwave bands (e.g., 9.2 μm), including the atmospheric window (Sokolik and Toon, 1999), with additional significant contributions from both the silt- and clay-sized





minerals (Fig. S19). But, its absorption at most bands (e.g., band 3: 15.87-20 μm) implemented in CAM5 is weak or
comparable with that of other minerals (Fig. S20). As a result, the perturbing analysis highly likely underestimated the
sensitivity of the longwave DRE to variations of the mineral contents and the uncertainty in the longwave DRE. Our
calculation neglecting dust scattering of longwave radiation shows that the global mean longwave DRE deviates from the
baseline by ±0.02 W m$^{-2}$, resulting in an uncertainty range of [+0.09, +0.13] W m$^{-2}$, with large values mainly found along the
"dust belt" (Fig. S21).


Previous studies have suggested that omitting longwave dust scattering results in an underestimate of the longwave DRE by
between ~23-51% (Sicard et al., 2014; Dufresne et al., 2002). The estimated deviation from the baseline in the longwave
DRE becomes ±0.03 W m$^{-2}$ due to perturbed parameters (e.g., imaginary complex refractive index for each mineral), if we
artificially augment the longwave DRE at the TOA by 51% attempting to include scattering effects following previous
studies (Miller et al., 2006; Biagio et al., 2020; Kok et al., 2017). This results in an estimate of the longwave DRE ranging
between [+0.14, +0.20] W m$^{-2}$. MONARCH simulates a longwave dust DRE of +0.17 W m$^{-2}$. This value is the same as in
CAM with the 51% augmentation. Adding the simulated longwave dust DRE from MONARCH to that of CAM thus leads to
little change in the longwave DRE uncertainty.

### 3.4 Net (sum of shortwave and longwave) direct radiative effect uncertainty

Our baseline simulation shows a net DRE warming of +0.04 W m$^{-2}$ (Fig. 14d), which is close to the estimate of -0.03 W m$^{-2}$
obtained by Di Biagio et al. (2020). The net DRE we estimate is strongly contrasted to the cooling effect as obtained by
AEROCOM (-0.5 W m$^{-2}$) and Kok et al. (-0.26 W m$^{-2}$, 2017). The longwave warming induced by both dust scattering
(augmentation by 51%) and absorption almost completely (longwave : shortwave ≈0.92 in absolute terms) offsets the
shortwave cooling at the TOA obtained in CAM5 with C1999, which is slightly larger than the longwave : shortwave ratio
range (0.23-0.88 in absolute terms) reported in previous studies (Kok et al., 2017; Di Biagio et al., 2020).

We estimate the range of the net dust DRE to be between [-0.23, +0.35] W m$^{-2}$ using CAM5 with both soil atlases being
considered. Therefore, dust has a probability of ~60% to warm the planet, a factor of 2.4 higher than the estimate of Kok et
al. (2017), who argued that there was a 25% chance that dust warms. The net dust DRE range becomes [-0.22, +0.34] W m$^{-2}$
if considering iron oxides only in the shortwave DRE calculation (longwave DRE is not totally insensitive to variations of
mineral amounts in our model). The uncertainty in the soil abundance of iron oxides, therefore, contributes ~97%
((0.34+0.22) / (0.35+0.23) × 100 = 97%) to the total uncertainty for CAM5. Thus, we identify iron oxides as the largest
uncertainty source and can be more important than dust burden or the imaginary refractive index of minerals.

The inclusion of multiple-models results into the abovementioned estimate yields the largest net DRE range of [-0.30, +0.36]
W m$^{-2}$ to date. The uncertainty range, to a certain extent, reflects the influence of different model treatments of parameters

(on e.g., size distribution, emission, transport, mixing states of minerals or dust with other species, and atmospheric processing), which is smaller than that of uncertainties in parameters we considered in CAM5. Using this estimate, soil mineral uncertainties account for ~85% ((0.34+0.22) / (0.36+0.30) × 100 ≈ 85%) of the total range in DRE calculated in this
study.

## 4 Conclusions

Iron oxides including hematite and goethite are the most important mineral absorbers at solar wavelengths (Sokolik and Toon, 1999; Claquin et al., 1999; Lafon et al., 2006; Balkanski et al., 2007; Formenti et al., 2014; Journet et al., 2014; Scanza et al., 2015; Li and Sokolik, 2018). Here, for the first time we perform comprehensive studies to address uncertainty
in dust DRE arising from the abundance of iron oxides in soil mineralogy atlases, C1999 and J2014. We estimate this uncertainty in DRE by using dust mineralogy-speciated climate models and focusing in particular on iron oxides with their known uncertainties in C1999 and J2014. Detailed sensitivity studies were performed using a perturbation analysis methodology on the eight different minerals and associated imaginary refractive indices along with dust AOD. Uncertainties in iron oxide content represent ~97% of the uncertainties estimated considering CAM only, and ~85% across multi-model
uncertainties.

While hematite is a more absorbing iron oxide that goethite, our results show that uncertainty in goethite in J2014 produces a larger uncertainty in the shortwave DRE estimate, even larger than the uncertainty caused by the hematite differences between C1999 and J2014. Given the volume averaging method used in the model to compute bulk aerosol optical
properties, despite J2014 being the latest soil map, its introduction does not improve CAM5 predictions of the observed DRE efficiency at the TOA over North Africa and downwind regions. While C1999 assumed that iron oxides are all in the form of hematite, our tests highlight the importance of distinguishing goethite from hematite for the shortwave DRE estimate. Otherwise, the model tends to underestimate dust warming at the TOA by ~56%.

Sensitivity studies in CAM5, which represents internally-mixed aerosol species within each mode, demonstrated that the shortwave dust DRE at the TOA is highly sensitive to estimates of the iron oxide atmospheric burden; iron oxides along with other minerals considered in this study have a negligible influence on the longwave DRE. As a consequence, the large uncertainty in the amount of hematite present in soils leads to an uncertainty up to 0.32 W m$^{-2}$ in the TOA shortwave DRE. We conclude that to estimate the shortwave DRE, the modelled fraction and speciation of iron oxides must be considered in
addition to parameters such as the size distribution and imaginary complex refractive index of minerals. When including the longwave warming in our model, there is about a 60% probability that mineral dust produces a net warming at TOA (Fig. 14).

The use of the volume averaging method to compute the bulk dust optical properties (e.g., complex refractive index) based on the dust mineral species probably overestimate absorption (Zhang et al., 2015; Li and Sokolik, 2018), leading to an artificial warming in CAM5 and CAM6. Our model very likely underestimates a large fraction of the coarse-mode dust particles (diameter >5 μm) according to a recent study (Adebiyi and Kok, 2020), and thus underestimates the dust warming effect. In addition, the transport of "giant" dust particles (diameter >20 μm) is still a representation issue that remains unsolved. Treatments of the "giant" dust particles as have been considered in previous studies (e.g., Di Biagio et al., 2020)

will continue with future studies. See detailed discussions about some other sources of the DRE uncertainty estimate in Appendix A. Even though they are not explicitly accounted for in the perturbation analysis in CAM, the influence of some of these remaining elements on the DRE may have been in part covered by using multiple models as reflected in the large model spread.

Considering that improving modelled mineralogical composition of dust is important to other disciplines or research subjects such as biogeochemistry and dust-cloud interactions, a new soil atlas with more accurate hematite soil distribution is required. New measurement methods are expected to produce such an atlas (Green et al., 2020). Incorporating this information will improve a model's ability to quantify and understand the DRE by mineral dust and its role in the Earth system.

**Data availability.**

Data will be available on the Cornell eCommons respository (https://doi.org/10.7298/wedj-jv65).

**Author contributions.**

NMM and LL designed the study with discussions with RLM, CPGP, PG, MK, DSH, OK, VO, and DP. LL performed CAM simulations, analysed multiple model results with comments from NMM, RLM, CPGP, DSH, and MK, and wrote the

manuscript with support from NMM, DSH, RLM, CPGP, MGA, MK, and PG; M.K. performed MONARCH simulations and analysed dust AOD and SSA in MONARCH; RLM performed ModelE2 simulations; PG performed GFDL simulations. DP performed line-by-line calculations. MGA, CPGP, and YB provided Journet soil atlases. JFK performed the mass extinction efficient calculation for non-spherical and spherical dust. All authors edited manuscript texts.

**Acknowledgements.**

This work was supported by the NASA EMIT project and the Earth Venture – Instrument program. We acknowledge high-performance computing resources provided by NCAR's Computational and Information Systems Laboratory. LLL, NMM,





and DSH was supported by the Atkinson Centre for a Sustainable Future. JFK acknowledges support from NSF grant 1552519. MK has received funding from the European Union's Horizon 2020 research and innovation programme under the Marie Skłodowska-Curie grant agreement No. 789630. CPGP and MG acknowledge support by the European Research Council (grant no. 773051, FRAGMENT), the AXA Research Fund, the Spanish Ministry of Science, Innovation and Universities (RYC-2015-18690 and CGL2017-88911-R), and PRACE for awarding access to MareNostrum at Barcelona Supercomputing Center to run MONARCH. RLM acknowledges support from the NASA Modeling, Analysis and Prediction Program (NNG14HH42I).

**Competing interests.**

The authors declare that they have no conflict of interest.

**Appendix A: other sources of uncertainty**

In this appendix, we compare the mineral speciation uncertainties to some of other major sources of dust DRE uncertainty. Our perturbation analysis has not explicitly accounted for all elements that are relevant to this estimate in CAM, which are discussed here.

Size distribution is known as an important parameter that strongly affects the dust DRE (Mahowald et al., 2014). The base shortwave DRE obtained in CAM5 based upon C1999 relies heavily on the aerosol size distribution employed in CAM5. The representation of the size distribution is an issue that remains as yet unsolved (Li et al., in prep). A single larger dust particle, typically has a higher absorption efficiency and lower scattering efficiency in the shortwave spectrum range. Therefore, even for the size-independent mineralogical composition, although the complex refractive index of each mineral does not depend on size (Sokolik et al., 1993; Sokolik and Toon, 1999), the SSA decreases steadily as the fraction of large-sized dust increases. Recent observations show significantly abundant coarse and even "giant" (diameter >20 μm) dust particles, over the Sahara and islands downwind (Johnson and Osborne, 2011; Ryder et al., 2013, 2019). Consequently, an aerosol cut-off diameter of 10 μm in CAM could bias our baseline towards more cooling, since coarse particles have shorter lifetimes and tend to absorb shortwave radiation more than fine particles (Kok et al., 2017; Granados-Muñoz et al., 2019). A recent study (Adebiyi and Kok, 2020) found that most models including ModelE2 and CESM significantly underestimate the fraction of dust particles with the diameter greater than 5 μm in the atmosphere compared to in situ measurements of dust size distributions compiled from publications. Because the dependence of SSA on composition is important only when the coarse fraction is low (Di Biagio et al., 2019; Ryder et al., 2013), the importance of iron oxides is probably overestimated here owing to missing a large fraction of coarse-mode dust by the models.





A major source of hematite is the Sahel (Hamilton et al., 2019; Scanza et al., 2015), whose emission is sensitive to the model dynamics and dust generation scheme, even though here the model wind is nudged towards MERRA. Even though the dust scheme used by CAM (Kok et al., 2014a) shows some improvements compared to DEAD in the model-observation
comparison (Hamilton et al., 2019; Kok et al., 2014b), there are still large uncertainties in representing surface soil conditions of dust source areas in global models. Despite the insensitivity of dust mass extinction efficiency to mineralogy, a new generation scheme that yields a different emission pattern could change the mass fraction of iron oxides of dust aerosol across the globe. This could modify the shortwave DRE, even with the same globally mean dust AOD.

Apart from the emission, many aspects of modelling dust transport (dry and wet deposition, dust-cloud interaction, and mixing states with other aerosols such as sulfate, black carbon, and sea salt) remain subject to large uncertainties. Most of them are related to uncertainties in 1) parameterizations of the dust cycle, as well as 2) the simulated meteorology propagating in part from the reanalysis products, to which that dust mobilization is sensitive. Most models, therefore, could not perfectly reproduce the observational dust distributions (Albani et al., 2014; Ginoux et al., 2001; Huneeus et al., 2011;
Mahowald et al., 2005). This is true because of the limited spatial coverage and temporal frequency of observational datasets, and their sampling bias with few measurements over remote regions. For instance, both CAM4 and CAM5 match dust deposition observations within a factor of 10 and overestimate it at some sites in Europe and Antarctica, while overestimating it over the South Pacific (Albani et al., 2014). Although a notable difference exits in the dust spatial distribution among the multiple models used in this study, it is possible that the simulated spatial distributions of dust
minerals do not bracket the full range of observations in dust plume extents or burdens, leaving out a part of uncertainty.

The ageing process (like e.g., heterogenous chemistry) of individual dust particles acts to alter their chemical composition. For example, high-level calcite-containing dust from e.g., parts of China and Saudi Arabia have been found to react with nitric acid and form a nitrate salt (Krueger et al., 2004). The salt compounds cause increased update of water vapor from the
atmosphere and thus growth of the particle size. As a result, compared to non-aged particles, aged dust is more efficiently removed by the wet and dry deposition, leading to a reduced dust burden and lifetime (Abdelkader et al., 2017). Growth of particle size by deliquescence also changes the optical properties. The importance of the atmospheric processing on changing physical-chemical properties dust aerosol depends on its mineralogy and transport path, which determine the species (e.g., secondary acids, ammonium) that accumulate on the dust surface (Sullivan et al., 2007). In contrast to the Asia dust case
(Krueger et al., 2004), optical properties and chemical composition of transported dust in Mediterranean from the Saharan Desert show negligible changes, despite mixing with pollution particles (Denjean et al., 2016). These processes, unfortunately, are still not well established.

Other relevant uncertainties for the DRE estimate that are not explicitly considered here include:1) the altitude of the dust
plume (Granados-Muñoz et al., 2019), especially its location with respect to clouds which can affect the DRE at both





shortwave and longwave bands (Huang et al., 2009); 2) representation of surface albedo; 3) mixing assumptions, two extreme states of which shown in CAM4 and CAM5, when in reality, the mixing state of dust minerals along with other species is somewhere in between; 4) nano-sized iron oxides that are commonly associated with clay minerals but are not represented in the CAM model; 5) hygroscopicity for each mineral which is assumed to be identical regardless of mineral

composition; and 6) the efficiency of transmitting fine-mode aerosols to coarse-model aerosols through particle coagulation.

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





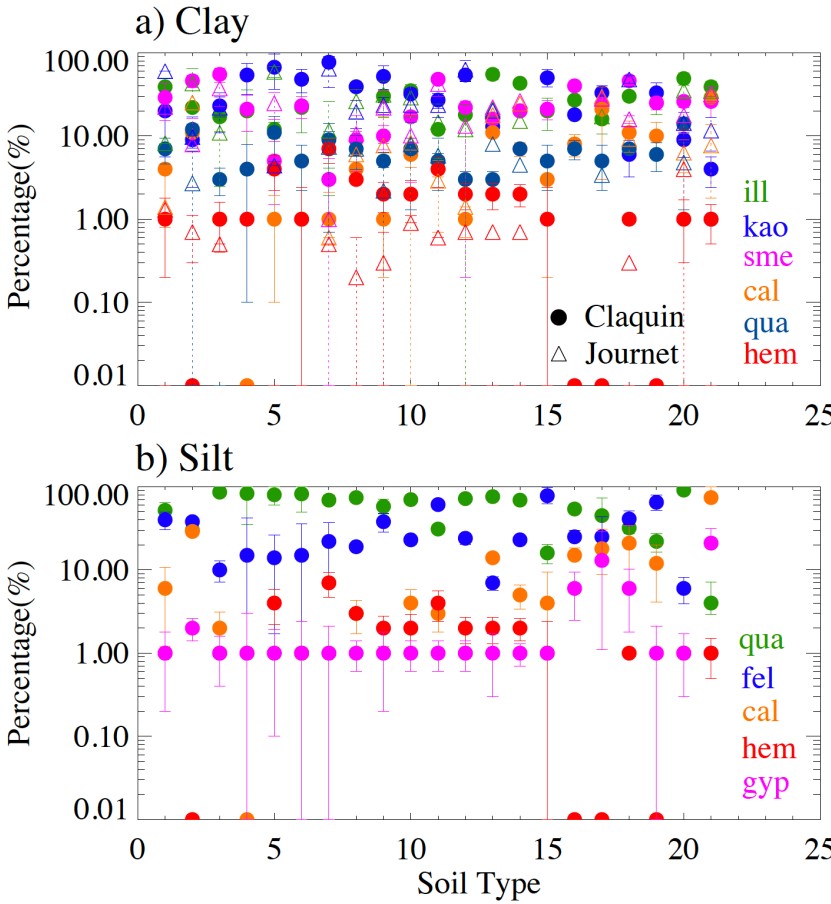


**Figure 1: Mean mineral percentage (C1999: colored filled dots; J2014: triangle) and associated uncertainty (error bars) in the clay- (a) and silt-sized (b) categories based on C1999 and J2014 for each soil type. X-axis labels from 1 to 21 corresponds to the first column of Table 2 of (Claquin et al., 1999) from top to bottom. Soil units used for comparison to C1999 data are listed in Table 3 of Journet et al. (2014) and are reordered here according to X-axis labels used for C1999 soil types.**




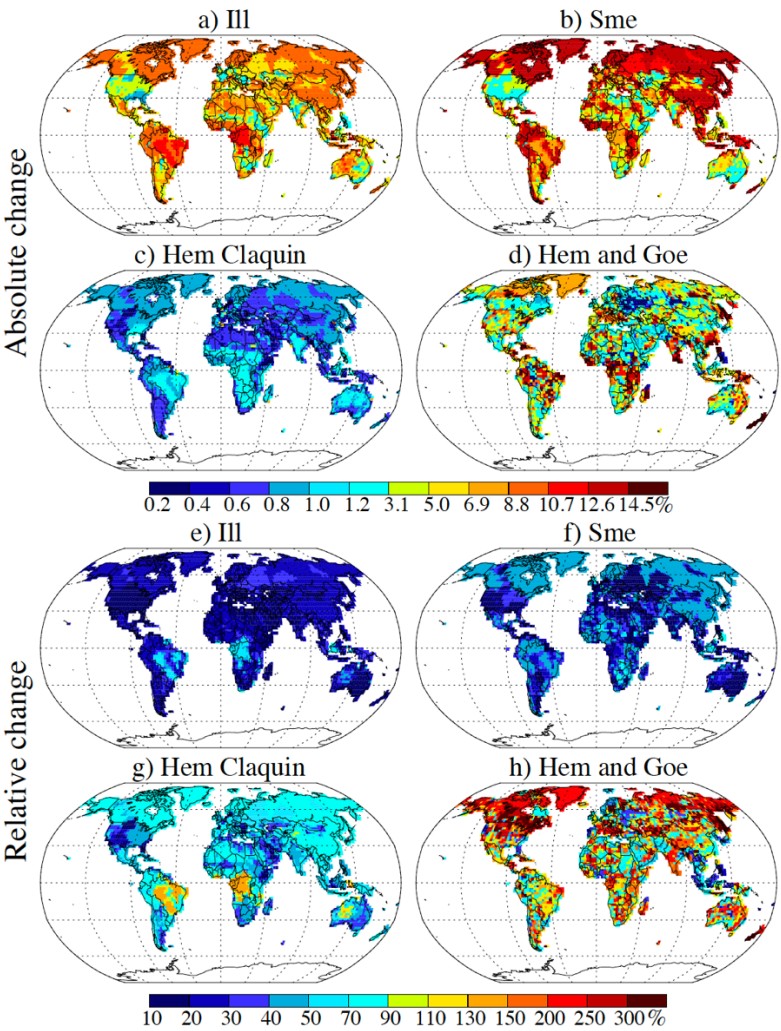

**Figure 2: Changes of soil concentration (fractional amount) of illite (ill), smectite (sme), hematite (hem), and goethite (goe) in the clay category. In a, b, c, and d, values are derived by subtracting high-bound minerals as shown in Figure 1 indicated by error bars for each grid cell from their base constructed following the method of Scanza et al. (2015) according to the mean mineralogy table (MMT) in C1999 (a, b, and c) and J2014 (d: hematite plus goethite) in CAM5. Similarly, e, f, g ,and h shows the relative change defined as (high bound-base)/(base)*100. The mean soil distribution of these minerals has been shown previously (Scanza et al., 2015; Perlwitz et al., 2015), and are repeated. Because of the limited information on mineral content in the silt-sized category, to create the global atlas for dust modelling showing the high and low bound of iron oxides, we applied to all soil units a constant standard deviation of goethite that is present for two soil units for which we have information.**



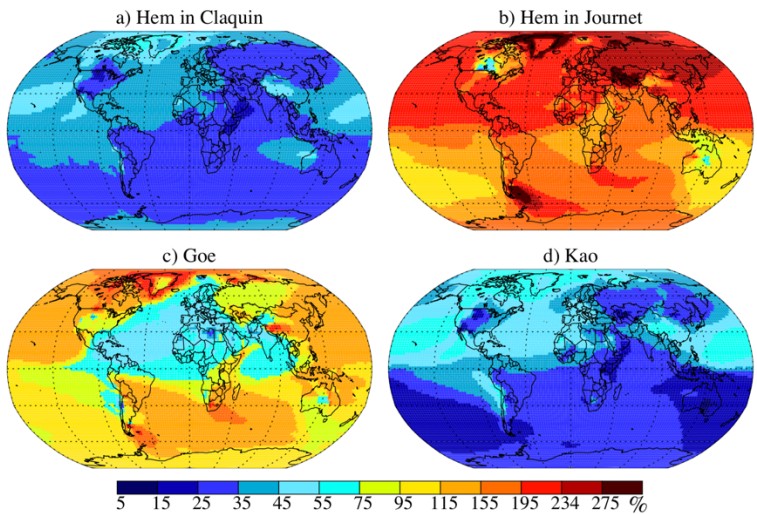


**Figure 3:** **Relative change (in percentage) of simulated mass fraction for hematite (hem) C1999 (a; in the clay-sized category) and J2014 (b), goethite (c, goe), and kaolinite (d, kao) in CAM5 from base to high bounds of their soil distribution. Relative change in percentage is calculated as (high bound-base)/(base)×100. The mean distributions have been shown previously (Scanza et al., 2015; Perlwitz et al., 2015), and are repeated.**


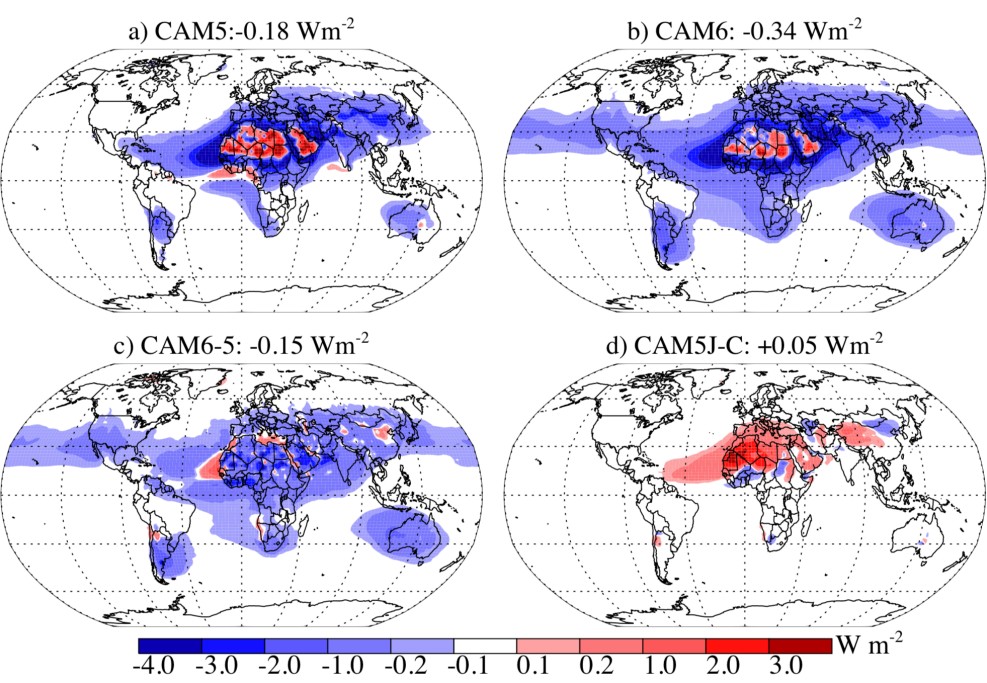




**Figure 4: Shortwave TOA DRE (W m⁻²) in CAM5 (a) and CAM6 (b) with C1999 (a,b, and c) and J2014 (d), and their differences (c and d) for 2007-2011. DRE in CAM6 was regridded onto CAM5 grids. Numbers in the title show global mean DRE (a and b) or difference: between CAM6 and CAM5 (c); between CAM5 with J2014 and with C1999 (d).**

**Figure 5: Global mean shortwave DRE by dust (a) and single scattering albedo at the 0.44-0.63 μm band (b) averaged over pixels where AOD$_{dust}$>0.5•AOD$_{total}$ following Scanza et al. (2015) in CAM5 for different cases in C1999 (first seven bars from the left) and J2014 (last bar from the right). Values associated with parameters other than iron oxides, imaginary complex refractive index, and dust AOD were derived from the "normalized" cases (see Section 2.3.1). Red dash lines indicate values obtained from the baseline simulation; blue dash lines denote values obtained from the simulation with J2014 distinguishing hematite from goethite; purple**





**dash lines are similar to blue ones but with identical optical properties between hematite and goethite. Bars: values associated with higher (in color) and lower bounds (dash with opposite signs to real values) of minerals, dust AOD, and imaginary complex refractive index. X-axis labels: Hem-hematite; Sme-smectite; Ill-illite; Kao-kaolinite; Cal-calcite; Qua-quartz; Fel-feldspar; Gyp-gypsum; DOD-dust AOD; Ima-Imaginary; J. iron oxide-iron oxides in J2014.**

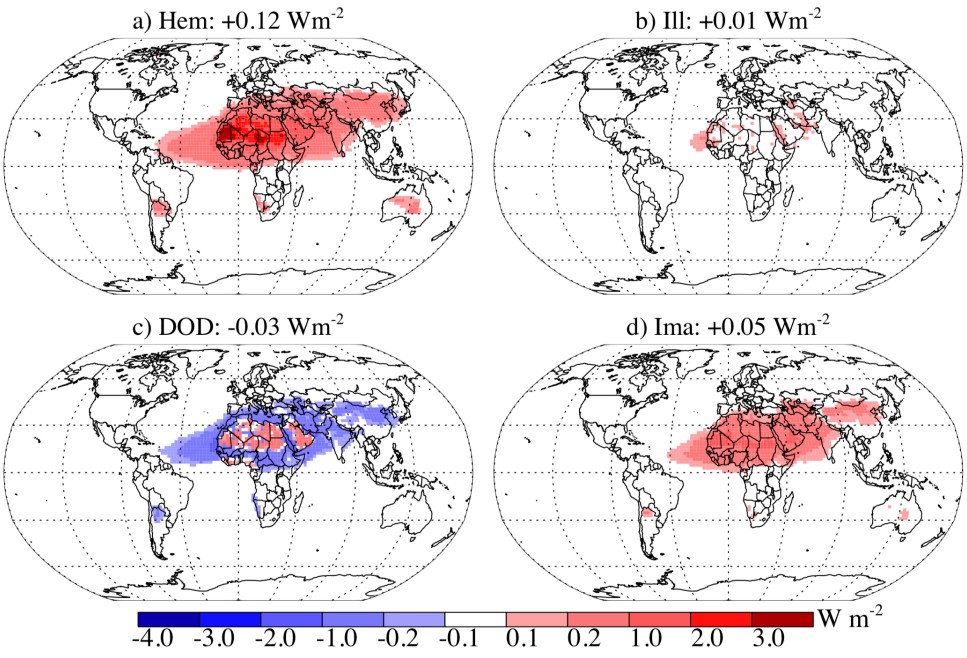

**Figure 6: Upper branch of uncertainty in TOA shortwave DRE W m$^{-2}$ induced by uncertainty in hematite (a, Hem), illite (b, Ill), dust AOD (c, DOD), and imaginary complex refractive index (d, Ima) in CAM5. All simulations used here are based on C1999. Numbers in the title denote global mean deviation from the baseline in CAM5. Values are calculated at each grid box as the difference between DRE from the high-bound soil mineralogy case and the baseline.**



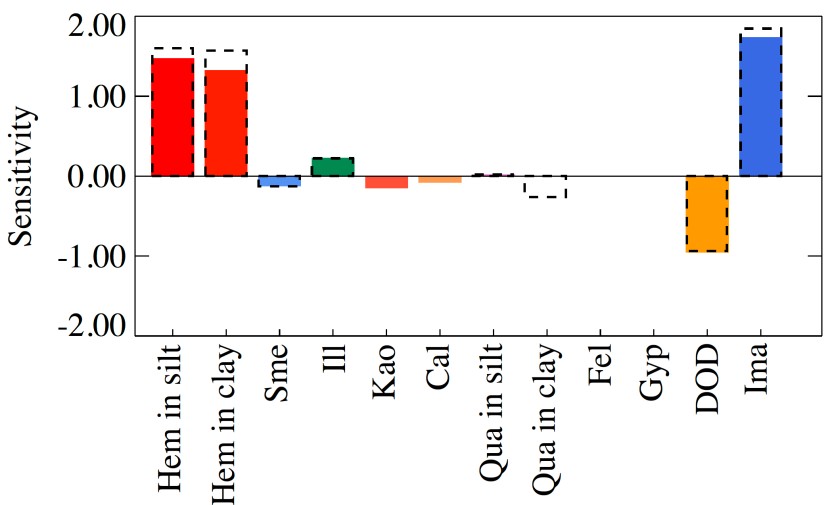

**Figure 7: Sensitivity parameter (unitless) of the shortwave DRE to simulated minerals (hematite, smectite, and illite), dust AOD, and the prescribed imaginary complex refractive indices within the known uncertainty in CAM5. The sensitivity is measured by the ratio of the relative change of shortwave DRE to that of parameters considered. Bars: values associated with higher (in color) and lower bounds (dash with opposite signs to real values) of minerals, dust AOD, and imaginary complex refractive index. X-axis labels: Hem-hematite; Sme-smectite; Ill-illite; Kao-kaolinite; Cal-calcite; Qua-quartz; Fel-feldspar; Gyp-gypsum; DOD-dust AOD; Ima-Imaginary; J. iron oxide-iron oxide in J2014. Sensitivity for parameters other than hematite, dust AOD, and imaginary complex refractive index, was derived from the "normalized" cases (see Section 2.3.1).**



**Figure 8: Upper branch of shortwave DRE uncertainty estimated considering all parameters (a and b) and iron oxides only (c and d) in CAM5 with the soil mineral distribution coming solely from C1999 (a and c) and both C1999 and J2014 (b and d). Numbers show the high (in the title) and low branches (inlet) of the global mean uncertainty estimated based on the global average shortwave DRE in individual cases (See Section 2.3.3). White color denotes values below 0.1 W m$^{-2}$.**

415





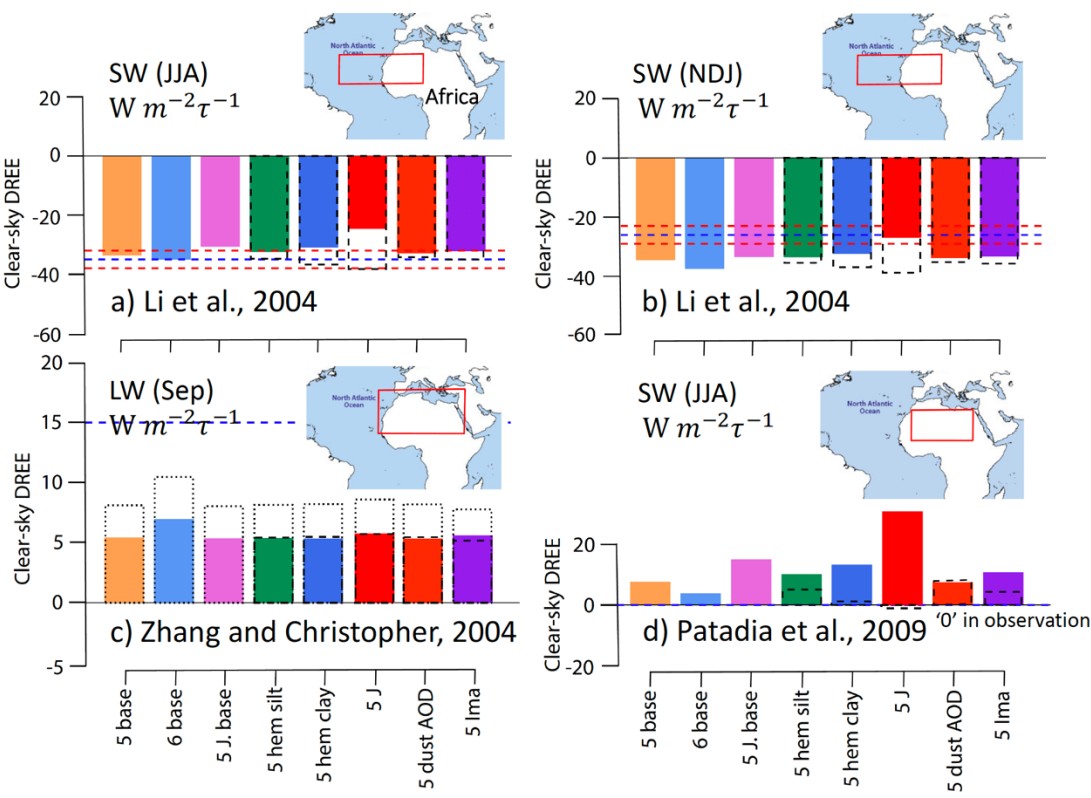

Figure 9: Comparison of clear-sky shortwave (SW) and longwave (LW) dust DRE efficiency (unit: W m$^{-2}$ $\tau^{-1}$) to observation at the top of the atmosphere (TOA). The model-observation comparison is in summer and winter over North Atlantic (a and b; JJA and NDJ; Li et al., 2004), September (c, Sep; Zhang and Christopher, 2004), and summer over North Africa (d; JJA; Patadia et al., 2009) for the longwave and shortwave spectral range, respectively. The DRE efficiency is calculated as a ratio of DRE under clear-sky conditions to simulated dust AOD (indicated by $\tau$). First three bars from the left: DRE efficiency calculated in CAM5 and CAM6 with mean soil data of J2014 and C1999; last five bars: values obtained from runs in CAM5 with high (in color) and low (dash) bounds. Horizonal blue lines denote observational mean, and two red dash lines in (a) and (b) denote uncertainty in the observations. Note zero SW dust DRE efficiency in the observations over North Africa in Summer (d) (Patadia et al., 2009). Inlet maps with the read box show the location where observational DRE efficiency are made and used for comparison. X-axis labels: 5 base-CAM5 with C1999; 6 base- CAM6 with C1999; 5 J. base- CAM5 with J2014; 5 dust AOD-CAM5 dust AOD; 5 Ima-CAM5 imaginary complex refractive index. Dot boxes indicate longwave DRE augmented by 51%.





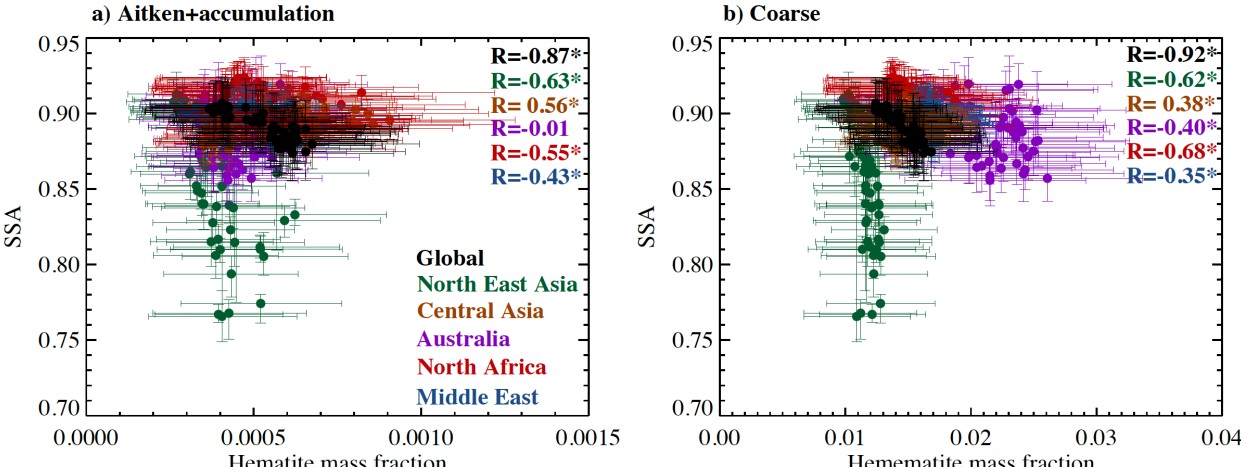

**Figure 10.** Single scattering albedo (SSA) at the 0.44-0.63 µm band versus the mass fraction of hematite in Aitken plus

1435 accumulation (a) and coarse (b) modes for different sub continental regions (Middle East, North Africa, Australia, Central Asia, North East Asia as indicated in the legend in color) and for global continents (in black). Simulations in CAM5 with C1999 for baseline, perturbed iron oxide mass fractions, dust AOD, and imaginary complex refractive index are used for analysis. Each point represents an area-average annual mean for each simulation. Pixels identified as ocean mask and having $AOD_{dust} \leq 0.5 \cdot AOD_{total}$ for land mask are removed for the regional analysis. Error bars indicate one standard deviation of the derived dust SSA and

1440 simulated hematite aerosol mass fraction in different modes. Also shown is correlation coefficient between the derived dust SSA and hematite aerosol mass fraction. Stars indicate that the correlation is statistically significant at the 95% confidence level.

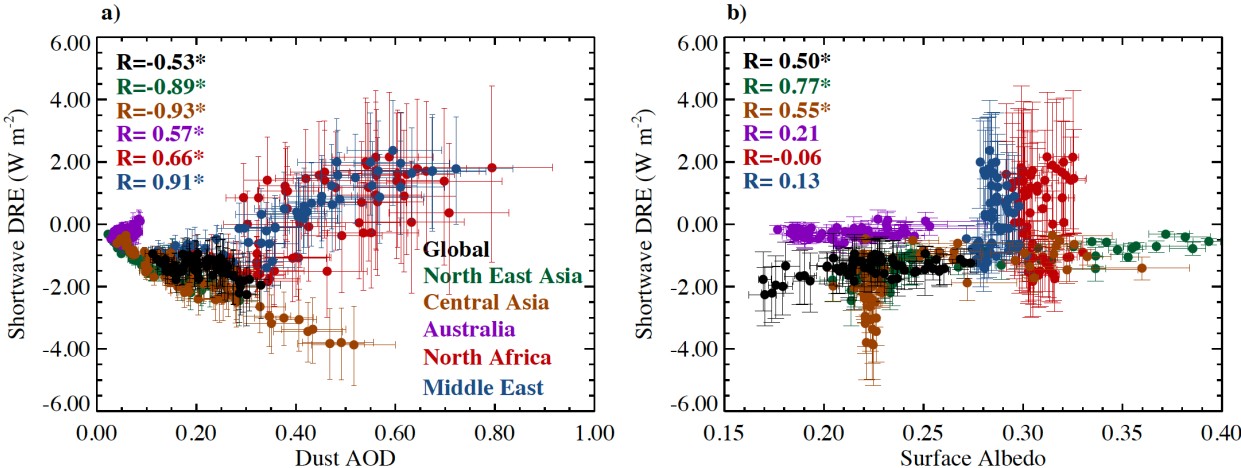

**Figure 11:** As in Figure 10 but for shortwave DRE versus dust AOD and surface albedo.



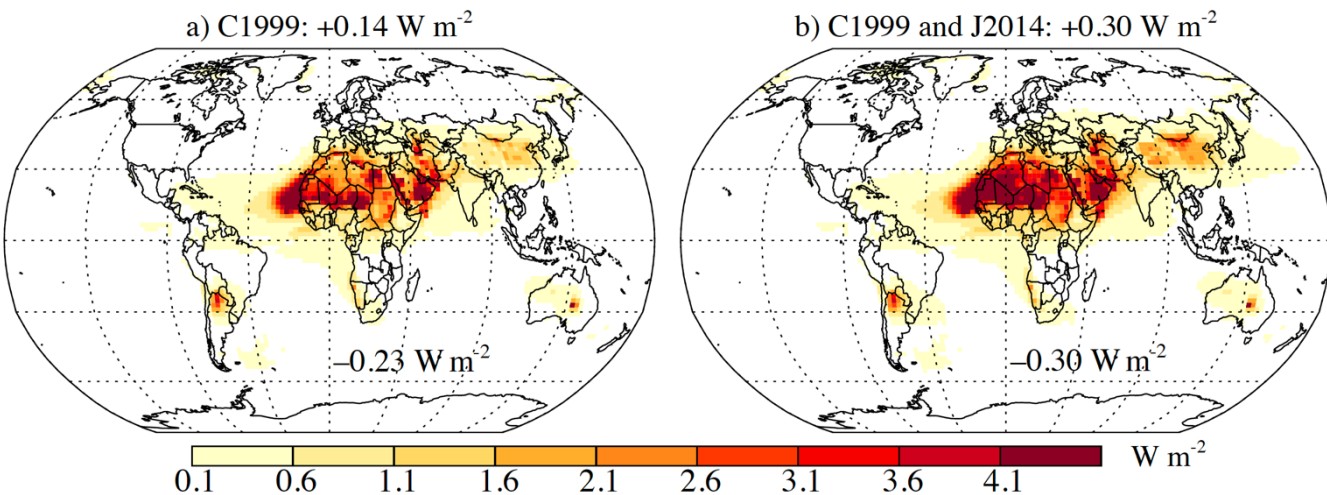

**Figure 12: As in Figure 8 but for the shortwave DRE uncertainty estimated based on a combination of five models (CAM5, CAM6, ModelE2, GFDL, MONARCH). Panel a) only includes soil distribution of minerals and their uncertainty in C1999 soil atlas. Panel b) further includes difference between C1999 and J2014, and uncertainty in J2014. Note that the shortwave DREs for ModelE2 and GFDL are obtained through regressions (see Section 2.3.4). Numbers show the high (in the title) and low branches (inlet) of the global mean uncertainty estimated based on the global average shortwave DRE in individual cases (See Section 2.3.3). White color denotes values below 0.1 W m$^{-2}$.**





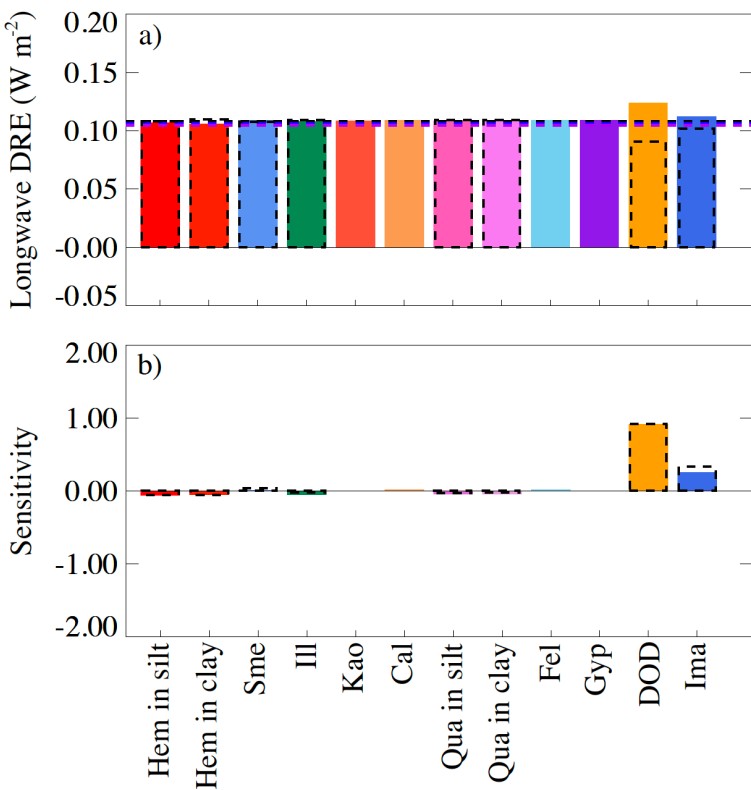

**Figure 13: Longwave DRE (a) and its sensitivity to minerals, dust AOD, and imaginary complex refractive index (b) in CAM5. In panel a), black lines indicate values obtained from the simulation with C1999; blue lines denote values obtained from the simulation with J2014 distinguishing hematite from goethite; purple lines are similar to blue ones but with identical optical properties between hematite and goethite. Bars: values associated with higher (in color) and lower limits (dash with opposite signs to real values) of minerals, dust AOD, and imaginary complex refractive index. X-axis labels: Hem-hematite; Sme-smectite; Ill-illite; Kao-kaolinite; Cal-calcite; Qua-quartz; Fel-feldspar; Gyp-gypsum; DOD-dust AOD; Ima-Imaginary.**



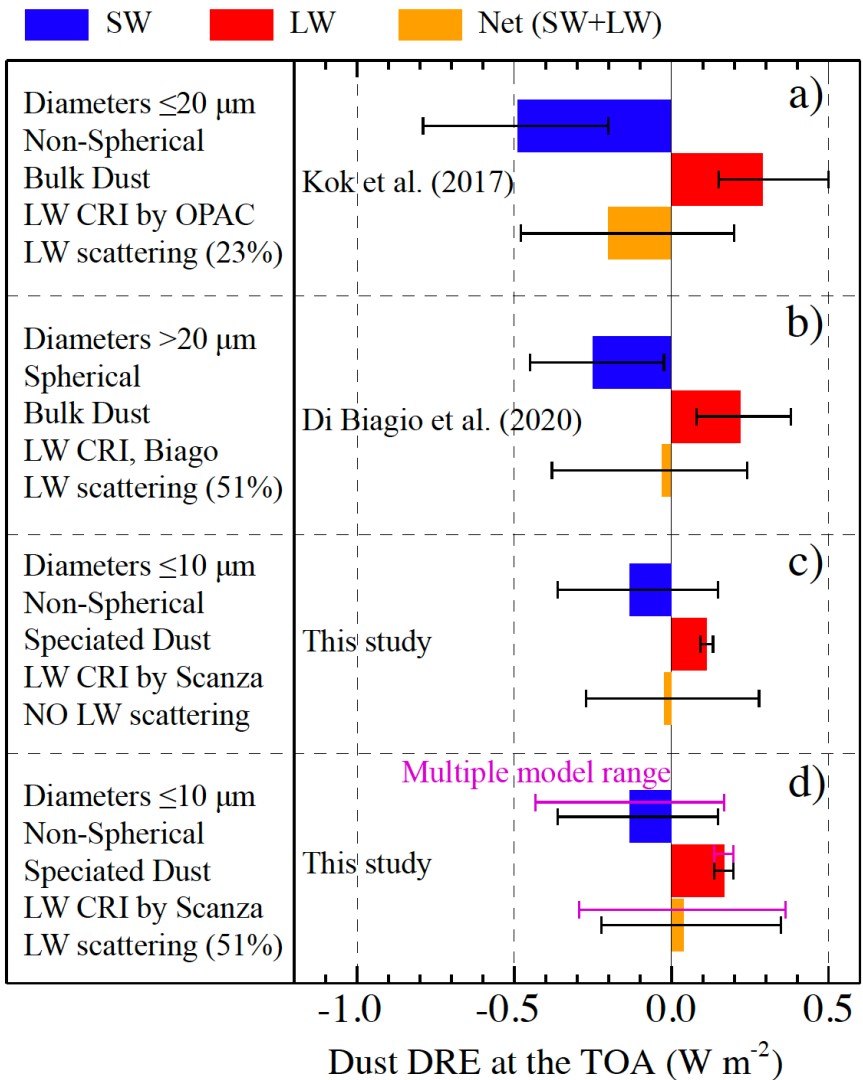

**Figure 14:** Comparison of global mean DRE at the top of the atmosphere (TOA) obtained by Kok et al. (2017) (a), Biago et al. (2019) (b), and this study (c and d). In panels c) and d), black error bars denote estimate based on CAM5 and CAM6 with mineral uncertainty in C1999 and J2014. Purple bars in panel d) represents estimate based on multiple models (CAM5, CAM6, ModelE2, GFDL, MONARCH with both soil maps), and the longwave DRE in CAM is scaled up by ~1.5. Note uncertainty of the longwave radiative effect is obtained based on CAM5, CAM6, and MONARCH. Texts to the left describe detailed information used for corresponding estimates. The description on this study applies to CAM5 and CAM6 only. GFDL also has a cut-off diameter of 10 μm. ModelE2 and MONARCH consider dust particles with the diameter up to 50 and 20 μm, respectively. Kok et al. (2017) utilized complex refractive index (CRI) from Optical properties of Aerosols and Clouds (Hess et al., 1998) or Volz (1973). Estimate in Biago et al. (2020) is based on CRI obtained from Biago et al. (2019). Speciated-dust model utilizes CRI of each mineral taken from Scanza et al. (2015). Dust optics in MONARCH are for LW; for SW optics, see texts.



**Table 1: Modal Aerosol Model (MAM) mode size parameters in CAM5 and CAM6**

| CAM6 | Modes | Geometric standard deviation | Geometric mean diameter (μm) |
|---|---|---|---|
|  | 1: accum | 1.6 | 0.11 |
|  | 2: aitken | 1.6 | 0.026 |
|  | 3: coarse | 1.2 | 0.90 |
|  | 4: primary | 1.6 | 0.05 |
| CAM5 | 1: accum | 1.6 | 0.11 |
|  | 2: aitken | 1.6 | 0.026 |
|  | 3: coarse | 1.8 | 2.0 |
|  | 4: primary | 1.6 | 0.020 |

**Table 2: List of experiments for the sensitivity test using CAMs (CAM4, CAM5, and CAM6), ModelE2, MONARCH, and GFDL with speciated (indicated by C1999 and J2014) and bulk dust (indicated by N/A in the "Soil maps" column). ModelE2, GFDL, and MONARCH results were regridded onto CAM5 grids, so, the resolution column does not reflect the original setting. Note hem-hematite; sme-smectite; ill-illite; Kao-kaolinite; cal-calcite; qua-quartz; fel-feldspar; gyp-gypsum; Ima-Imaginary; LW-longwave; SW-shortwave.**

| Models | Configuration | Descriptions | Soil maps | Resolutions | Optics |
|---|---|---|---|---|---|
| CAM4 | FSDBAM | Baseline | C1999 | 1.9° × 2.5° | Scanza et al. (2015) |
| CAM5[a,b] | FC5 | Claquin baseline | C1999 | 1.9° × 2.5° | Scanza et al. (2015) |
| CAM6[b,c] | F2000climo | Baseline | C1999 | 1.9° × 2.5° | Scanza et al. (2015) |
| CAM5[b,c] | FC5 | Journet baseline | J2014 | 1.9° × 2.5° | Scanza et al. (2015)* |
| CAM5[b,c] | FC5 | Same hem and goe | J2014 | 1.9° × 2.5° | Scanza et al. (2015)* |
| CAM5[b,c] | FC5 | High iron oxide | J2014 | 1.9° × 2.5° | Scanza et al. (2015)* |
| CAM5[b,c] | FC5 | Low iron oxide | J2014 | 1.9° × 2.5° | Scanza et al. (2015)* |
| CAM5[a,d] | FC5 | High ill clay | C1999 | 1.9° × 2.5° | Scanza et al. (2015) |
| CAM5[a,d] | FC5 | Low ill clay | C1999 | 1.9° × 2.5° | Scanza et al. (2015) |
| CAM5[a,d] | FC5 | High sme clay | C1999 | 1.9° × 2.5° | Scanza et al. (2015) |
| CAM5[a,d] | FC5 | Low sme clay | C1999 | 1.9° × 2.5° | Scanza et al. (2015) |
| CAM5[a,d] | FC5 | High qua silt | C1999 | 1.9° × 2.5° | Scanza et al. (2015) |
| CAM5[a,d] | FC5 | Low qua silt | C1999 | 1.9° × 2.5° | Scanza et al. (2015) |
| CAM5[a,d] | FC5 | High qua clay | C1999 | 1.9° × 2.5° | Scanza et al. (2015) |
| CAM5[a,d] | FC5 | Low qua clay | C1999 | 1.9° × 2.5° | Scanza et al. (2015) |
| CAM5[a,d] | FC5 | High cal clay | C1999 | 1.9° × 2.5° | Scanza et al. (2015) |





| CAM5[a,d] | FC5 | High kao clay | C1999 | $1.9° \times 2.5°$ | Scanza et al. (2015) |
|---|---|---|---|---|---|
| CAM5[a,d] | FC5 | High gyp silt | C1999 | $1.9° \times 2.5°$ | Scanza et al. (2015) |
| CAM5[a,d] | FC5 | High fel silt | C1999 | $1.9° \times 2.5°$ | Scanza et al. (2015) |
| CAM5[a] | FC5 | Aitken hem removed | C1999 | $1.9° \times 2.5°$ | Scanza et al. (2015) |
| CAM4 | FSDBAM | High hem clay | C1999 | $1.9° \times 2.5°$ | Scanza et al. (2015) |
| CAM5[a,b] | FC5 | High hem clay | C1999 | $1.9° \times 2.5°$ | Scanza et al. (2015) |
| CAM5[a,b] | FC5 | Low hem clay | C1999 | $1.9° \times 2.5°$ | Scanza et al. (2015) |
| CAM5[a,b] | FC5 | Low hem silt | C1999 | $1.9° \times 2.5°$ | Scanza et al. (2015) |
| CAM5[a,b] | FC5 | Low hem silt | C1999 | $1.9° \times 2.5°$ | Scanza et al. (2015) |
| CAM5[a,b] | FC5 | Low dust AOD | C1999 | $1.9° \times 2.5°$ | Scanza et al. (2015) |
| CAM5[a,b] | FC5 | Low dust AOD | C1999 | $1.9° \times 2.5°$ | Scanza et al. (2015) |
| CAM5[a,b] | FC5 | High Ima | C1999 | $1.9° \times 2.5°$ | Scanza et al. (2015) |
| CAM5[a,b] | FC5 | Low Ima | C1999 | $1.9° \times 2.5°$ | Scanza et al. (2015) |
| ModelE2 | N/A | Baseline | C1999 | $1.9° \times 2.5°$ | N/A |
| MONARCH | N/A | Baseline | N/A | $1.9° \times 2.5°$ | LW: OPAC; SW: see texts |
| GFDL | N/A | Baseline | N/A | $1.9° \times 2.5°$ | N/A |

**\* Assumed optical properties for goethite**

**[a] and [b] model simulations with and without the bug, respectively**

**[c] model simulations without bug and without considering the dust shape effect**

**[d] a scaling factor applied to the calculated DRE**

**Table 3: Dust AOD and burdens (Tg) in CAM4, CAM5 with C1999 and J2014, CAM6 with C1999 with hematite coming solely**
**from the clay-sized category, ModelE2 with C1999, GFDL, and MONARCH. Note differences in the global mean dust SSA**
**calculation between CAMs and MONARCH: in CAM, the global mean dust SSA was derived from the simulated SSA for total**
**aerosols at the 0.44-0.63 μm band by retaining only pixels with $AOD_{dust}>0.5 \cdot AOD_{total}$ in the calculation following Scanza et al.**
**(2015); in MONARCH, the global mean SSA was calculated based on the simulated SSA at the 0.44-0.63 μm band for pure dust**
**aerosol; in GFDL, the global mean SSA was calculated based on the simulated SSA at the 0.50-0.60 μm band for pure dust aerosol.**

| Models | Dust aerosol mass (Tg) | Dust AOD | SSA |
|---|---|---|---|
| CAM4 | 26 | 0.032 | 0.96 |
| CAM5 (C1999) | 25 | 0.031 | 0.89 |
| CAM5 (J2014) | 25 | 0.030 | 0.87 |
| CAM6 (C1999) | 24 | 0.030 | 0.90 |
| ModelE2 | 24 | N/A | N/A |
| GFDL | 16 | 0.020 | 0.96 |





| MONARCH | 24 | 0.027 | 0.92 |
|---|---|---|---|

**N/A: no data**

**Table 4: Simulated mineral mass fraction, and fractional absolute and relative changes (in percentage, %) of mineral mass fraction from mean to the high bound in global average.**

| Cases | Mean | Low | Absolute change | Relative change (%) | High | Absolute change | Relative change (%) |
|---|---|---|---|---|---|---|---|
| hematite in clay | 1.65 | 1.09 | 0.56 | 33.94 | 2.22 | 0.57 | 34.55 |
| hematite in silt | 1.65 | 1.43 | 0.22 | 13.33 | 1.87 | 0.22 | 13.33 |
| illite | 27.12 | 22.20 | 4.92 | 18.14 | 32.05 | 4.93 | 18.18 |
| kaolinite | 16.55 | N/A | N/A | N/A | 22.36 | 5.81 | 35.11 |
| calcite | 6.95 | N/A | N/A | N/A | 8.34 | 1.39 | 20.00 |
| Quartz in clay | 21.60 | 20.40 | 1.20 | 5.56 | 22.80 | 1.20 | 5.56 |
| Quartz in silt | 21.60 | 19.70 | 1.90 | 8.80 | 24.00 | 2.40 | 11.11 |
| Feldspar | 7.50 | 5.89 | 1.61 | 21.47 | 9.25 | 1.75 | 23.33 |
| Gypsum | 0.54 | N/A | N/A | N/A | 0.86 | 0.32 | 59.26 |

**N/A: no data**

**Table 5: Global mean single scattering albedo (SSA) at the 0.44-0.63 μm band, and DRE in shortwave (SW), longwave (LW) spectrum and their sum (Net) for different cases in CAM5 and CAM6. Dust AOD for all cases is approximately 0.03 except "high dust AOD" (~0.035) and "low dust AOD" (~0.025). Values in the last (right) four rows are obtained in CAM5 using J2014 with different (baseline) and identical optical properties for hematite and goethite (same hem and goe). See descriptions for the case** 505 **name in Table 2.**

| Case names | SSA[a] | SSA$_1$[b] | SSA$_2$[b] | SW[a] | SW | LW[a] | LW | Net[a] | Net |
|---|---|---|---|---|---|---|---|---|---|
| Claquin baseline | 0.895 | 0.892 | 0.889 | -0.142 | -0.184[b] | 0.084 | 0.108[b] | -0.058 | -0.076[b] |
| High hem silt | 0.891 | 0.884 | 0.880 | -0.116 | -0.148[b] | 0.083 | 0.107[b] | -0.033 | -0.041[b] |
| Low hem silt | 0.902 | 0.900 | 0.899 | -0.169 | -0.222[b] | 0.084 | 0.109[b] | -0.085 | -0.114[b] |
| High hem clay | 0.883 | 0.873 | 0.868 | -0.082 | -0.100[b] | 0.083 | 0.106[b] | 0.001 | 0.006[b] |
| Low hem clay | 0.909 | 0.912 | 0.913 | -0.211 | -0.282[b] | 0.085 | 0.110[b] | -0.126 | -0.172[b] |
| High dust AOD | 0.896 | 0.892 | 0.889 | -0.164 | -0.213[b] | 0.096 | 0.124[b] | -0.068 | -0.089[b] |
| Low dust AOD | 0.894 | 0.891 | 0.890 | -0.120 | -0.155[b] | 0.071 | 0.091[b] | -0.049 | -0.064[b] |
| High sme clay | 0.895 | N/A | N/A | -0.147 | -0.191[c] | 0.084 | 0.109[c] | -0.063 | 0.081[c] |
| Low sme clay | 0.896 | N/A | N/A | -0.137 | -0.178[c] | 0.083 | 0.108[c] | -0.054 | -0.070[c] |
| High ill clay | 0.896 | N/A | N/A | -0.136 | -0.177[c] | 0.083 | 0.108[c] | -0.053 | -0.069[c] |





| | | | | | | | | | |
|---|---|---|---|---|---|---|---|---|---|
| Low ill clay | 0.895 | N/A | N/A | -0.148 | -0.192[c] | 0.084 | 0.109[c] | -0.064 | -0.083[c] |
| Low Ima | 0.903 | 0.902 | 0.901 | -0.181 | -0.238[b] | 0.079 | 0.102[b] | -0.102 | -0.136[b] |
| High Ima | 0.889 | 0.882 | 0.878 | -0.105 | -0.133[b] | 0.087 | 0.112[b] | -0.018 | -0.021[b] |
| High qua clay | 0.895 | N/A | N/A | -0.144 | -0.187[c] | 0.084 | 0.109[c] | -0.061 | -0.079[c] |
| Low qua clay | 0.895 | N/A | N/A | -0.140 | -0.182[c] | 0.084 | 0.109[c] | -0.056 | -0.073[c] |
| High qua silt | 0.896 | N/A | N/A | -0.142 | -0.184[c] | 0.083 | 0.108[c] | -0.058 | -0.076[c] |
| Low qua silt | 0.896 | N/A | N/A | -0.142 | -0.185[c] | 0.084 | 0.109[c] | -0.058 | -0.076[c] |
| High gyp silt | 0.895 | N/A | N/A | -0.142 | -0.185[c] | 0.084 | 0.109[c] | -0.059 | -0.076[c] |
| High kao clay | 0.896 | N/A | N/A | -0.150 | -0.195[c] | 0.084 | 0.109[c] | -0.066 | -0.086[c] |
| High fel silt | 0.895 | N/A | N/A | -0.142 | -0.185[c] | 0.084 | 0.109[c] | -0.058 | -0.076[c] |
| High cal clay | 0.895 | N/A | N/A | -0.144 | -0.188[c] | 0.084 | 0.109[c] | -0.061 | -0.079[c] |
| CAM6 base | 0.900 | 0.900 | 0.903 | -0.440 | -0.337[b] | 0.195 | 0.144[b] | -0.246 | -0.194[b] |
| Journet baseline | 0.880 | 0.874 | 0.867 | -0.254 | -0.136[b] | 0.156 | 0.106[b] | -0.099 | -0.030[b] |
| Same hem and goe | 0.864 | 0.857 | 0.847 | -0.136 | -0.045[b] | 0.153 | 0.105[b] | 0.017 | 0.059[b] |
| High iron oxide | 0.847 | 0.817 | 0.800 | -0.091 | 0.122[b] | 0.151 | 0.100[b] | 0.060 | 0.106[b] |
| Low iron oxide | 0.903 | 0.923 | 0.925 | -0.320 | -0.326[b] | 0.143 | 0.099[b] | -0.178 | -0.116[b] |

[a] **obtained in models runs with incorrect mass specification for dust AOD calculation (see Section 2.3.1)**

[b] **obtained in models runs with correct mass specification for dust AOD calculation**

[c] **"normalized" cases (see Section 2.3.1)**

[1] **dust SSA calculated based upon pixels that have $AOD_{dust} > 0.5 \cdot AOD_{total}$ (dust fractional threshold: 0.5)**

[2] **dust SSA calculated with a higher dust fractional threshold (0.8) than in "1".**

**N/A: no data**