# Peer review of "Quantifying the range of the dust direct radiative effect due to source mineralogy uncertainty"

_Atmospheric Chemistry and Physics, 2020_

## Referee Comment (RC1) · Anonymous Referee #1 · 18 Sep 2020

Review of "Quantifying the range of the dust direct radiative effect due to source mineralogy uncertainty," by Li et al., submitted to Atmospheric Chemistry and Physics.

This paper discusses the direct radiative forcing of mineral dust accounting for uncertainties in soil mineral abundance, composition, and optical properties.

Whereas, the paper appears comprehensive and includes many references, it doesn't discuss possibly the most related paper, that of Jacobson (2001). That study simulated the solar and thermal-IR global direct radiative forcing of several individual soil dust components, namely iron oxide, aluminum oxide, silicon dioxide, calcium carbonate, magnesium carbonate, potassium carbonate, and sodium carbonate, as well as soil dust as a whole. Of possible relevance, the present paper does not mention anywhere the role of absorption by aluminum in soil dust particles, Figure 4k of Jacobson (2001)

indicates that aluminum may be a strong absorber in soil dust particles.

Ideally, the authors would include aluminum in their calculations. I realize this could result in having to redo their entire calculations. At a minimum, the authors need to discuss this omission and the potential impact on results. The authors also should mention that study in the context of its findings with regards to individual chemical components and the overall soil dust radiative impact in the solar and thermal-IR.

Introduction. An additional impact of absorption by soil dust components is to contribute to cloud burn-off (Jacobson, 2012). Please discuss briefly.

Methods. "Two datasets currently exist. . ." Please clarify that FAO (1995) includes world soil data at 10 km resolution, and includes soil composition (SiO2, CaCO3, CaSO4, Fe2O3, Illite, Kaolinite, Smectite, Feldspars) in each data cell.

What is the source of solar- and thermal-infrared refractive index data for each chemical? It would be useful to see a plot of real and imaginary refractive indices of absorbing components versus wavelength.

What is missing is a comparison of aerosol absorption optical depth with global satellite data. This would give a better idea of the realism of the results here.

Figure S1. Please specify the wavelength range of "shortwave" radiation assumed. Also, please define "high-bound hematite" and "high-bound dust" in the figure caption.

Figure S2. Please provide the source of the data in the figure caption.

References

FAO, Soil Map of the World, Land and Water Dev. Div., Rome, Italy, 1995.

Jacobson, M.Z., Global direct radiative forcing due to multicomponent anthropogenic and natural aerosols, J. Geophys. Res., 106, 1551-1568, 2001.

Jacobson, M.Z., Investigating cloud absorption effects: Global absorption properties of

black carbon, tar balls, and soil dust in clouds and aerosols, J. Geophys. Res., 117, D06205, doi:10.1029/2011JD017218, 2012.

---

## Referee Comment (RC2) · Anonymous Referee #2 · 5 Oct 2020

The manuscript describes the uncertainty of the top-of-the-atmosphere dust direct radiative forcing due to current uncertainties in the surface soil mineralogy. Especially the importance of iron oxides is high-lighted.

The manuscript is well-organized and include detailed description of the results. It is recommended for publication after consideration of the minor comments below.

Comments

- **Page 2, lines 45-46**: The sentence starting with "These two..." is unclear. Please rewrite.

[Figure]

- **Page 2, line 54**: The net dust DRE is given. May you also include numbers for other DREs so the reader can have an idea about the dust DRE magnitude in relation to other processes?

- **Page 7, line 193**: Remove "in the model".

- **Page 10, line 297**: Remove "." after ")".

- **Page 10, line 297**: What is a "simple double call"?

- **Page 15, line 465**: Should $\Delta F_{upp}$ be $\Delta F_{hig}$?

- **Page 21, line 650**: Change "7a" to "7".

- **Page 24, line 752**: The region of Australia is marked with a star in Fig 11a, indicating that the result is statistically significant. This is the opposite of what is said in the text.

- **Page 25, line 784**: Change "African" to "Africa".

- **Page 26, lines 797-808**: The radiative parameterization is discussed. However, it is not clear that the same radiative transfer solution method is used in both the line-by-line and the parameterized calculations. If the methods are different, for example if one of the methods use more streams, this will potentially cause significant differences.

- **Page 28, line 867**: Change "that goethite" to "than goethite".

- **Page 30, line 922**: Change "of other" to "of the other".

- **Page 31, lines 956-958**: Unclear sentence (too many whiles?), please rewrite.

- **Page 31, line 958**: Change '"exits" to "exists".

- **Page 45, line 1370**: What is meant by "and are repeated"?

- **Page 46, line 1388; page 47, line 1392; page 48, line 1404; page 52, line 1456; Fig. S13**: The acronym DOD is used in the captions and figures. It is not mentioned anywhere in the text. In the text you use $AOD_{dust}$. Please use only one notation for the dust AOD throughout the manuscript.

- **Page 53, lines 1474-1475**: Unclear sentence, please rewrite.

- **Table S2**: Please provide full reference to the CESM User Guide.

---

## Referee Comment (RC3) · Anonymous Referee #1 · 17 Oct 2020

For indices of refraction of Corundum (Al2O3) from the UV to thermal-IR, please see Koike et al., Icarus, 114, 203-214, 1995. The imaginary refractive index at 0.4 microns is 0.043 and at 0.5 microns is 0.0382 and at 0.6 microns is 0.0367 from ISAS (Table A1). These values are ∼1/4th those for iron oxide (e.g., which is around 0.15 at 0.5 microns), thus definitely important. The authors are correct that it will depend on concentration and whether Al2O3 can be a surrogate for Al in a mixture. These are important issues to mention.

---

## Author Comment (AC1) · 17 Oct 2020

We thank the reviewer much for the comments and criticism on this work. We have made changes to the manuscript where necessary. Text from the manuscript is quoted with double quotation marks. Please see figures/tables we created and cited here as Fig. R/Table R in the supplement material.

COMMENT

Whereas, the paper appears comprehensive and includes many references, it doesn't discuss possibly the most related paper, that of Jacobson (2001). That study simulated the solar and thermal-IR global direct radiative forcing of several individual soil dust components, namely iron oxide, aluminum oxide, silicon dioxide, calcium carbonate,

magnesium carbonate, potassium carbonate, and sodium carbonate, as well as soil dust as a whole. Of possible relevance, the present paper does not mention anywhere the role of absorption by aluminum in soil dust particles, Figure 4k of Jacobson (2001) indicates that aluminum may be a strong absorber in soil dust particles.

Ideally, the authors would include aluminum in their calculations. I realize this could result in having to redo their entire calculations. At a minimum, the authors need to discuss this omission and the potential impact on results. The authors also should mention that study in the context of its findings with regards to individual chemical components and the overall soil dust radiative impact in the solar and thermal-IR.

RESPONSE

Aluminum oxide is known as corundum in the mineralogy terminology. However, Jacobson (2001) is expressing aluminum, silicon and iron in minerals as component oxides. (Chemical analyses of minerals are customarily reported as weight percent's of component oxides.) For example, the mineral feldspar that we referred to in the manuscript is a group of rock-forming tectosilicate minerals. Being one of the alkali feldspar (a solid solution between two endmembers, potassium feldspar and albite), (Na0.87K0.13)Al1Si3O8 is equivalent to the combination of component oxides shown in Table R1 (please note that we are using the figure/table number of our own as can be seen in each figure/table).

Therefore, our methodology cannot be directly compared to that of Jacobson (2001). What we model here are dust mineral species (illite, kaolinite, smectite, feldspar, iron oxides, etc.). In fact, these minerals could be equally expressed in terms of the weight percent of component oxides such as silicon dioxide, aluminum oxide, and iron oxide, as termed by Jacobson (2001).

Besides this, while in nature there may be contributions of pure aluminum oxide (i.e., the mineral corundum) in dust, its contribution is very small, and it is not reported in any of the available soil mineralogical atlases that we used to compute dust aerosol

mineralogy. As discussed below the absorption of aluminum oxide is negligible:

1) The absorption of pure crystal aluminum oxide that is present in dust aerosol particles would be negligible, in particular at the UV and visible bands, or at least significantly lower than that of iron oxides. We are not quite sure about what aluminum oxide refers to and the source of the optical constants used in Jacobson (2001). According to Table 2 of Toon and Pollack (1976), the imaginary complex refractive index (ImCRI) of pure crystal aluminum oxide approximates to an order of -7 at the UV and visible bands (as shown in Fig. R1), values much lower than that of iron oxides (an order between -1 and 1 at the visible Band 10 and UV Band 11 in CAM5/6). We obtained the wavelength-dependent CRI for dust minerals simulated by our models from Scanza et al. (2015) (please see supplement, Section S2 of that study), who compiled the CRI data from in-lab measurements. Toon and Pollack (1987) derived the CRI of pure crystal aluminum oxide based on measurements made at room temperature or 1200 celsius degree. As temperature increases, the absorption coefficient (related to the ImCRI) for the pure crystal aluminum oxide can increase by dozens of times at the visible bands. But measurements at 3 $\mu$m, for example, show that the absorption coefficient does not substantially increase until temperature reaches 2000 K or above (Bityukov and Petrov, 2013) - a value well in excess of the upper limit of the Earth's atmospheric temperature.

2) The absorption by pure crystal aluminum oxide at infrared (IR) bands may not be as strong as minerals like quartz and feldspar that we considered in the manuscript for longwave dust direct radiative effect (DRE). For example, recently, Di Biagio et al. (2017) found that all the main absorption peaks at IR bands are contributions from either clays at 9.6 $\mu$m, quartz at 9.2 and 12.5-12.9 $\mu$m, kaolinite at 10.9 $\mu$m, calcite at 7.0 $\mu$m and 11.4 $\mu$m, or feldspars at 8.7 $\mu$m according to their in-situ measurements of dust spectral extinction coefficients. These measurements used soil samples selected from 137 available samples to be representative of the diversity of sources from arid and semi-arid regions at the global scale. The authors did not find contributions from

aluminum oxide to the main absorption peaks at IR bands.

3) Currently, single scattering albedo (SSA) is believed to be the most important parameter that determines dust DRE at the top of the atmosphere. Previous studies (e.g., Engelbrecht et al., 2016; Moosmuller et al., 2012) and our results all show a clear relationship between iron oxide content (mass fraction of total dust) and the resultant SSA, (most evident at 0.405 $\mu$m in Engelbrecht et al., 2016; at 0.405 $\mu$m and 0.870 $\mu$m in Moosmuller et al., 2012; at the band centered at 0.530 $\mu$m in our manuscript). To our best knowledge, no publications exist showing a clear relationship between SSA and aluminum oxide content at either shortwave or longwave bands.

4) As stated in the manuscript, we normalized the mineral fractions, and our model setup probably underestimates the sensitivity of longwave DRE to the mineral content variation. Because of these, even for quartz and feldspar, our calculation suggests they do not have considerable impacts on quantifying longwave DRE uncertainty. Moreover, the amount of the mineral corundum in the dust aerosol should be tiny in nature, considering the total amount of Al2O3 that can also come from a variety of other minerals (Table R1) we considered in the manuscript in dust is small, ranging between 0-15% (Figure 18b of Engelbrecht et al., 2016, which expressed the amount of Al in dust as Al2O3), in general, compared to quartz and feldspar. Consequently, the impact of omitting the mineral corundum, again whose absorption of longwave radiation is weak compared to quartz and feldspar, on the estimated longwave DRE range would be highly likely negligible as well.

Besides above discussions, we constructed a test in CAM5 with C1999 using the CRI of aluminum oxide reported by Toon and Pollack (1987) (Table 2 of that study and the data is shown in Fig. R1). Since the spatial distribution and uncertainty of corundum in the surface soil are unknown, we assume here they are the same as quartz in the silt-sized category. The amount (Fig. 1b of the preprint and Fig. 1e of Scanza et al. 2015) of high-bound quartz (the upper limit of the 95% confidence interval of quartz in the corresponding soil category) is higher than that of hematite in either the clayor silt-sized category; so is the absolute deviation of high-bound quartz from the base (Fig. 1b of the preprint); both the amount and the absolute deviation, thus, should be much higher than those of aluminum oxide or corundum in nature, the amount of which is tiny (please see point 4 above). Our assumption on the spatial distribution and uncertainty of aluminum oxide, thus, would highly likely exaggerate the impact of aluminum oxide on the estimated DRE uncertainty range. Despite the exaggeration, as can be seen in Fig. R2, the global mean deviation resulting from the assumed aluminum oxide uncertainty is still small, -0.003 W m-2 and +0.003 W m-2, for shortwave (Fig. R2a) and longwave (Fig. R2b with no 51% augment) DRE, respectively. With the above assumptions, we conclude that the impact of aluminum oxide on the global DRE uncertainty range, is highly likely much smaller in amplitude than that caused by hematite (Fig. 5a of the preprint). This is also true spatially, if one compares Fig. R2a to Fig. 6a (preprint) for shortwave DRE (longwave dust DRE insensitive to variation of the mineral content within the uncertainty range).

Therefore, further inclusion of aluminum oxide (or pure crystal aluminum oxide) into our model would unlikely lead to a substantially change to the DRE range we estimated. Note again that what we modeled are minerals, and these minerals could be equally expressed in terms of the weight percent of component oxides such as silicon dioxide, aluminum oxide, and iron oxide (Table R1), as termed by Jacobson (2001).

COMMENT

Introduction. An additional impact of absorption by soil dust components is to contribute to cloud burn-off (Jacobson, 2012). Please discuss briefly.

RESPONSE

This is good point. Cloud burn-off occurs when dust and cloud are co-located due to absorption of shortwave radiation by dust which will increase diabatic heating and enhance cloud evaporation. Though the cloud burn-off effect of dust is outside the topic of this study, it's good to mention in the introduction for completeness. Following

the reviewer's suggestion, we cited the work of Jacobson (2012) and modified relevant text in the introduction accordingly as follows:

"Dust aerosol (here defined as soil particles suspended in the atmosphere) perturbs the radiative energy balance directly by scattering and absorbing shortwave and long-wave radiation known as the aerosol-radiation interaction (Boucher et al., 2013) and indirectly by changing the cloud albedo and lifetime by acting as cloud condensation nuclei (CCN) and ice nuclei (IN) (Nenes et al., 2014) and by increasing diabatic heating in the atmosphere and evaporating cloud (Hansen et al., 1997; Bollasina et al., 2008; Jacobson, 2012) known as the aerosol-cloud interaction (Boucher et al., 2013)."

COMMENT

Methods. "Two datasets currently exist. . ." Please clarify that FAO (1995) includes world soil data at 10 km resolution, and includes soil composition (SiO2, CaCO3, CaSO4, Fe2O3, Illite, Kaolinite, Smectite, Feldspars) in each data cell.

RESPONSE

When creating soil mineralogy map, we did not take information about soil composition from FAO but utilized its soil legend and location information to map soil atlases from C1999 and J2014. Please see the second paragraph of Section 2.1 in main text.

COMMENT

What is the source of solar- and thermal-infrared refractive index data for each chemical? It would be useful to see a plot of real and imaginary refractive indices of absorbing components versus wavelength.

RESPONSE

We took the refractive index data from Scanza et al. (2015), who collected and reported the high-resolution data in their supplement.

Following the Reviewer's suggestion, we created a new figure (Fig. R3) showing

the CRI of each mineral against the shortwave and longwave bands implemented to CAM5/6. We show the CRI at each CAM5/6 band rather than the original high-resolution wavelength reported by Scanza et al., (2015), because the former is more directly linked to our dust DRE calculation.

Correspondingly, in the revised manuscript, we added new text like below introducing the data source, as well as the newly added figure.

"The refractive index of each mineral for each band implemented in CAM is derived from Scanza et al. (2015) and shown in Fig. R3 for CAM5/6."

Note the figure number "R3" would be changed in the text to reflect the real number it should be in the revised manuscript; the same applies to following text for the case that is in the same situation.

COMMENT

What is missing is a comparison of aerosol absorption optical depth with global satellite data. This would give a better idea of the realism of the results here.

RESPONSE

The model performance on simulating minerals and dust loading has been thoroughly evaluated by previous studies: CAMs in Scanza et al. (2015) for the mineralogy version, Kok et al. (2014a,b) for the new dust emission scheme; GISS ModelE2 in Perlwitz et al. (2015a,b); MONARCH in Pérez et al., (2011). The model-tuning method has been detailed in Albani et al. (2014). These references had already been cited in the preprint. So, we don't include them in the reference list here.

In the manuscript submitted, we had already compared the calculated radiative efficiency to observations near/over one of the major dust source regions, illustrated for North Africa on Fig. 9 (preprint). We also have pointed out in the methodology section that the volume averaging method we employed to compute the bulk dust optical properties is overestimating dust radiative absorption. Some other uncertainty sources

which we did not explicitly account for have also been given in Appendix A of the preprint. All these would show the readers how our models may bias the results.

However, to account for the reviewer's comments, we compared the results of the baseline simulation with measurements on the dust surface concentration/deposition and with an integrated dust aerosol optical depth (DOD) obtained by Ridley et al. (2016). We also compared the absorbing AOD (AAOD) from our models with that derived from AErosol RObotic NETwork (AERONET) measurements. Please see the details in the paragraphs that follow.

1) We plotted up a new figure (Fig. R4) to evaluate our model performance by comparing the simulated dust surface concentration and deposition flux with station-based measurements as in Albani et al. (2014). Please see detailed descriptions on the observational data from that study. Here, we show dust concentration/deposition from the baseline simulation only. For other cases, similar results were yielded and thus not shown, as DOD is insensitive to mineral content variation at least within their uncertainties (generally a small perturbation to the total dust amount). So, a retuning procedure for experiment cases except for high- and low-bound DOD is unnecessary, and thus both the surface dust concentration and deposition remain almost unchanged.

2) We added a comparison (Fig. R5) of simulated DOD with that obtained by integrating DOD from MODIS, MISR, AERONET, and model ensembles over 15 regions (x-axis labels of Fig. R5) as in Ridley et al. (2016).

3) In the result section (Section 3.2.2.2), to give another vision on our model performance, we added a comparison of the AAOD calculated from cases other than the "normalized" ones in Table R2 with that derived from AERONET measurements. There are no SSA constrains made in Ridley et al., (2016) for the 15 regions. SSA solely based on satellite retrievals are very likely subject to large uncertainty (Samset et al., 2018) for a large portion of areas with no station-based measurements available for calibration. Therefore, we did not compare with AAOD from satellite retrievals

but only utilized the DOD constrains as one of the proxies for evaluation of the model performance.

Per 1) and 2) above, we added the following text in Section 2.3.1:

"The baseline model fairly well reproduced the magnitude of dust concentration and deposition at the bottom model layer compared to station-based measurements (see Albani et al., 2014 for detailed descriptions) (Fig. R4; correlation: R2=0.88, and 0.83, for the surface dust concentration and deposition flux, respectively, which are statistically significant at the 95% confidence level). Particularly over regions near the dust source, such as North Africa, the model fairly well agrees with observations, despite a more smoothing spatial distribution of those dust proxies in the simulation. Comparing with the seasonal DOD averaged over 15 regions obtained by Ridley et al., (2016), the baseline simulation appears to show an overestimate in general near dust source regions and fairly well reproduced seasonal cycle (Fig. R5) from the climatological side. Periods for the simulation (2007-2011) and DOD constrain (2004-2008) do not well coincide. Despite the inconsistency in period, this overestimate of DOD close to the source is probably not totally an artifact, considering that to match DOD of 0.03 the global tuning of the model tends to emit more dust to compensate unduly strong deposition during transport. For the other cases, the simulated dust cycle is similarly comparable with observations and thus is not shown. The similarity of the simulated dust cycle among the different cases except those for high- and low-bound DOD is because DOD is insensitive to the variation of the mineral content at least within the mineral's uncertainty range, which is generally a small perturbation to the total dust amount. Therefore, a retuning procedure for experiment cases except for high- and low-bound DOD is unnecessary, and the simulated dust concentration and deposition, thus, remain almost unchanged."

Per 3), We revised Section 3.2.2.2 of the manuscript as follows:

a) changed the title to:

"Model to observation comparison: clear-sky radiative effect efficiency and absorbing aerosol optical depth"

b) interpreted new results concisely as below and shown in that section as well:

"The predicted absorbing AOD (AAOD) is well within one standard deviation ($\sigma$) of AERONET observations in all the cases except CAM6 with high-bound iron oxides in J2014 (Table R2). However, over the AERONET sites, CAM5/6 systematically undershoot observational AOD and with simulated values outside mean$\pm\sigma$ of the observation. The coincidence between predicted and observational AAOD accidently occurs, because, meanwhile, CAM5/6 overestimates the dust absorption of radiation near the 0.55 $\mu$m band with the simulated SSA systematically below the observation. It is likely that the overestimated radiation absorption is due to the use of the volume averaging method to compute the optical properties of bulk dust from those of the minerals. However, we cannot exclude the possibility of the contamination in dust over the selected sites by other absorbing aerosols like the black carbon. Moreover, the method used to filter out the AEROENT sites where dust aerosol does not dominate over other aerosols in terms of the optical depth (DOD no greater than 0.5xAOD) relies on the accuracy in the simulated DOD and non-dust AOD. Consequently, a mismatch that potentially exists between simulated and observational DOD and non-dust AOD may cause the comparison less meaningful. There are no SSA constrains made in Ridley et al., (2016) for the 15 regions. Thus, a comparison on the AAOD is unachievable. We did not compare the modeled AAOD with that from satellite observations, because available AAOD solely based on satellite retrievals are very likely subject to large uncertainty (Samset et al., 2018) for a large portion of areas with no station-based measurements available for calibration."

COMMENT

Figure S1. Please specify the wavelength range of "shortwave" radiation assumed. Also, please define "high-bound hematite" and "high-bound dust" in the figure caption.

RESPONSE

We specified the short and long wavelength ranges in the caption of Fig. S1 as follows.

"Short wavelength represents Bands 1-14 (band center range: 0.23-3.46 $\mu$m excluding the broad Band 14 centered at 8.02 $\mu$m), and long wavelength Bands 15-30 (band center: 3.46-514.29 $\mu$m) implemented in CAM5/6. These definitions apply to the whole rest of this supplement without further notice."

We defined "high-bound hematite" and "high-bound dust AOD" as well in that caption:

"For the case with the high-bound hematite in the clay-sized category, the model was configured to use the soil mineralogy atlas containing the upper limit of the 95% confidence interval of hematite in the corresponding category. The high-bound DOD and the other cases with high-bound/low-bound terms in this supplement are similarly defined."

Similarly, in the main text, for clarity, we

1) specified the short and long wavelength range: "The radiative flux at each vertical model layer, at 19 (band center range: 0.22-4.36 $\mu$m) and 14 (band center range: 0.23-3.46 $\mu$m excluding the broad Band 14 centered at 8.02 $\mu$m) shortwave bands (for CAM4 and CAM5/CAM6, respectively), and 16 longwave bands (band center range: 3.46-514.29 $\mu$m), . . ."

2) defined the "high-bound" term: "the high-bound hematite in the clay-sized category (a case with which the model utilized the soil mineralogy atlas that contains the upper limit of the 95% confidence interval of hematite in the corresponding category; the other high-bound or low-bound cases are similarly defined) . . ."

COMMENT

Figure S2. Please provide the source of the data in the figure caption.

RESPONSE

Added "in J2014" to the caption of Fig. S2.

COMMENT: references from the reviewer

FAO, Soil Map of the World, Land and Water Dev. Div., Rome, Italy, 1995.

Jacobson, M.Z., Global direct radiative forcing due to multicomponent anthropogenic and natural aerosols, J. Geophys. Res., 106, 1551-1568, 2001.

Jacobson, M.Z., Investigating cloud absorption effects: Global absorption properties of black carbon, tar balls, and soil dust in clouds and aerosols, J. Geophys. Res., 117, D06205, doi:10.1029/2011JD017218, 2012.

RESPONSE

Relevant references cited.
* * *
References
* * *
The references that had already been cited in the preprint are not included here.

Di Biagio, C., Formenti, P., Balkanski, Y., Caponi, L., Cazaunau, M., Pangui, E., Journet, E., Nowak, S., Caquineau, S., Andreae O, M., Kandler, K., Saeed, T., Piketh, S., Seibert, D., Williams, E. and Doussin, J. F. C.: Global scale variability of the mineral dust long-wave refractive index: A new dataset of in situ measurements for climate modeling and remote sensing, Atmos. Chem. Phys., 17(3), 1901–1929, doi:10.5194/acp-17-1901-2017, 2017.

Bityukov, V. K. and Petrov, V. A.: Absorption Coefficient of Molten Aluminum Oxide in Semitransparent Spectral Range, Appl. Phys. Res., 5(1), 51–71, doi:10.5539/apr.v5n1p51, 2013.

Bollasina, M., Nigam, S. and Lau, K. M.: Absorbing aerosols and summer monsoon

evolution over South Asia: An observational portrayal, J. Clim., 21(13), 3221–3239, doi:10.1175/2007JCLI2094.1, 2008.

Boucher, O., Randall, D., Artaxo, P., Bretherton, C., Feingold, G., Forster, P., Kerminen, V.-M., Kondo, Y., Liao, H., Lohman, U., Rasch, P., Satheesh, S., Sherwood, S., Stevens, B., Zhang, X.-Y., Lohmann, U., Rasch, P., Satheesh, S., Sherwood, S., Stevens, B. and Zhang, X.-Y.: Clouds and Aerosols, in Climate Change 2013: The Physical Science Basis. Contribution of Working Group I to the Fifth Assessment Report of the Intergovernmental Panel on Climate Change, edited by V. B. Stocker, T.F., D. Qin, G.-K. Plattner, M. Tignor, S.K. Allen, J. Boschung, A. Nauels, Y. Xia and P. M. Midgley, pp. 573–657, Cambridge University Press, United Kingdom., 2013.

Engelbrecht, J. P., Moosmüller, H., Pincock, S., Jayanty, R. K. M., Lersch, T. and Casuccio, G.: Technical note: Mineralogical, chemical, morphological, and optical interrelationships of mineral dust re-suspensions, Atmos. Chem. Phys., 16(17), 10809–10830, doi:10.5194/acp-16-10809-2016, 2016.

Hansen, J., Sato, M. and Ruedy, R.: Radiative forcing and climate response, J. Geophys. Res. Atmos., 102(D6), 6831–6864, doi:doi:10.1029/96JD03436, 1997. Holben, B. N., Eck, T. ., Slutsker, I., Tanre, D., Buis, J. P., Setzer, A., Vermote, E., Reagan, J. A., Kaufman, Y. J., Nakajima, T., Lavenu, F., Jankowiak, I. and Smirnov, A.: AERONET—A Federated Instrument Network and Data Archive for Aerosol Characterization, Remote Sens. Environ., 66, 1–16, doi:10.1007/BF03174421, 1998.

Jacobson, M. Z.: Investigating cloud absorption effects: Global absorption properties of black carbon, tar balls, and soil dust in clouds and aerosols, J. Geophys. Res. Atmos., 117(6), 1–25, doi:10.1029/2011JD017218, 2012.

Moosmuller, H., Engelbrecht, J. P., Skiba, M., Frey, G., Chakrabarty, R. K. and Arnott, W. P.: Single scattering albedo of fine mineral dust aerosols controlled by iron concentration, J. Geophsyical Res., 2006, 2004–2008, doi:10.1029/2011JD016909, 2012.

Samset, B. H., Stjern, C. W., Andrews, E., Kahn, R. A., Myhre, G., Schulz, M. and Schuster, G. L.: Aerosol Absorption: Progress Towards Global and Regional Constraints, Curr. Clim. Chang. Reports, 4(2), 65–83, doi:10.1007/s40641-018-0091-4, 2018.

Toon, O. B., Pollack, J. B. and Khare, B. N.: The optical constants of several atmospheric aerosol species: ammonium sulfate, aluminum oxide, and sodium chloride, J. Geophys. Res., 81(33), 5733–5748, doi:10.1029/JC081i033p05733, 1976.

Please also note the supplement to this comment:
https://acp.copernicus.org/preprints/acp-2020-547/acp-2020-547-AC1-supplement.pdf

[Figure]

**Supplement:**

*Supplement of*

**Response to Referee 1**

**Longlei Li et al.**

[ll859@cornell.edu](mailto:ll859@cornell.edu)

The copyright of individual parts of the supplement might differ from the CC BY 4.0 License.

**Table R1.** Component oxides that an alkali feldspar such as $(Na_{0.87}K_{0.13})Al_1Si_3O_8$ consists of.

| Oxide | Wt% | MolWt oxide | Moles oxide | Moles cation | Moles oxygen | Moles cations for 8O |
|-------|-----|-------------|-------------|--------------|--------------|----------------------|
| $SiO_2$ | 68.2 | 60.086 | 1.135 | 1.135 | 2.2701 | 2.9997 |
| $Al_2O_3$ | 19.29 | 101.963 | 0.1892 | 0.3784 | 0.5676 | 1.0001 |
| $Na_2O$ | 10.2 | 61.9796 | 0.1646 | 0.3291 | 0.1646 | 0.8699 |
| $K_2O$ | 2.32 | 94.2037 | 0.0246 | 0.0493 | 0.0246 | 0.1311 |

[Figure]

**Figure R1.** Real (a) and imaginary (b) complex refractive index (CRI) of Corundum (Cor) and hematite (Hem) for shortwave (blue shading) and longwave (green shading) bands (band centers shown as x-axis labels) implemented into CAM5/6. CRI values were derived for each band with original data from Scanza et al. (2015). Vertical dash lines indicate the shortwave Band 10 centered at 0.53 µm at which dust aerosol optical depth (DOD) and single scattering albedo were calculated (see Table 5 of the preprint) in CAM5/6. Note the band centered at 8.02 µm (leftmost) is broad with the boundaries of 3.84 and 12.20 µm. This broad band has been included in the model as shortwave bands by model developers.

[Figure]

**Figure R2:** Upper branch of uncertainty in shortwave (SW; a,) and longwave (LW; b; with no 51% augment) dust direct radiative effect W m$^{-2}$ at the top of the atmosphere induced by uncertainty in aluminum oxide with the assumption that the spatial distribution and uncertainty are the same as quartz in the silt-sized category. Simulations were constructed using C1999. We replaced the optical properties of quartz by those of aluminum oxide (Fig. R1) with the complex refractive index taken from Table 2 of Toon and Pollack (1987). Numbers in the title denote global mean deviation from the "baseline" (new experiment in which we also replaced quartz in the silt-sized category with aluminum oxide) in CAM5.

[Figure]

**Figure R3.** Real (a) and imaginary (b) complex refractive index (CRI) of each mineral for shortwave (blue shading) and longwave (green shading) bands (band centers shown as x-axis labels) implemented into CAM5/6. CRI values were derived for each band with original data taken from Scanza et al. (2015). The imaginary CRI of goethite was assumed to be half of hematite with the same spectral shape, while the real part of goethite is assumed to be identical as that of hematite. Vertical dash lines indicate the shortwave Band 10 centered at 0.53 μm at which dust aerosol optical depth and single scattering albedo for CAM5/6 were calculated (see Table 5 of the preprint). Note the band centered at 8.02 μm (leftmost) is broad with the low and high boundaries of 3.84 and 12.20 μm, respectively. This broad band has been included in the model as shortwave bands by model developers.

[Figure]

**Figure R4.** Comparison of simulated (the baseline case; see text for details) dust surface concentration, and deposition with observations. Also shown is correlation (both passed 95% statistically significant tests) between modeling and observations over sub-domains as indicated by texts in color. The dash lines in (b) and (d) represent 10:1 (upper left) and 1:10 (bottom right) lines.

[Figure]

**Figure R5.** Comparison of seasonally resolved dust aerosol optical depth (DOD) from the baseline simulation (blue) over 15 regions with that (brown) obtained in Ridley et al., (2016) who bias-corrected satellite-based retrievals from the Moderate Resolution Imaging Spectroradiometer (MODIS) and the Multi-angle Imaging Radiometer (MISR) using AErosol RObotic NETwork (AERONET) measurements and a model ensembles (see Ridley et al. 2016 for details). The shading area shows an example that the model greatly overestimated DOD compared to observations. Error bars represent the standard deviation.

**Table R2.** The climatologically mean total aerosol optical depth (AOD), absorbing aerosol optical depth (AAOD), and single scattering albedo (SSA) at 0.55 µm for AERONET (first portion) and at Band 10 centered at 0.53 µm for CAM5/6 (second portion). Values from CAM5/6 with J2014 (J) and/or C1999 (c) were obtained by averaging modeled AOD, absorbing AOD (AAOD), and SSA over the grid box nearest to the AERONET sites (e.g., Holben et al., 1998) where DOD>0.5•AOD (DOD represents dust AOD). Values in parenthesis show the standard deviation of AOD, AAOD, and SSA. Other notations: C(J)_bse: the baseline simulation with C1999 (J2014) ; J_Hig(Low) and J_Low: high(low)-bound of iron oxides in J2014, respectively; C_H(L)HemClay(Silt): high(low)-bound (see text for explanations) hematite in the clay(silt)-sized category; C_H(L)DOD: high(low)-bound DOD; C_H(L)Ima: high(low)-bound imaginary complex refractive index of minerals.

|                | AOD           | AAOD          | SSA           |
|----------------|---------------|---------------|---------------|
| AERONET        | 0.383(0.115)  | 0.046(0.011)  | 0.923(0.013)  |
| CAM6           | 0.209(0.057)  | 0.035(0.011)  | 0.899(0.008)  |
| CAM5C_bse      | 0.205(0.066)  | 0.039(0.011)  | 0.891(0.010)  |
| CAM5J_bse      | 0.205(0.065)  | 0.046(0.016)  | 0.875(0.006)  |
| CAM5J_Hig      | 0.202(0.063)  | 0.062(0.023)  | 0.837(0.010)  |
| CAM5J_Low      | 0.196(0.061)  | 0.031(0.008)  | 0.907(0.007)  |
| CAM5C_HHemClay | 0.205(0.065)  | 0.044(0.013)  | 0.879(0.010)  |
| CAM5C_LHemClay | 0.206(0.066)  | 0.034(0.009)  | 0.903(0.010)  |
| CAM5C_HHemSilt | 0.205(0.065)  | 0.041(0.012)  | 0.886(0.010)  |
| CAM5C_LHemSilt | 0.206(0.066)  | 0.048(0.010)  | 0.896(0.010)  |
| CAM5C_HDOD     | 0.228(0.075)  | 0.043(0.012)  | 0.891(0.010)  |
| CAM5C_LDOD     | 0.184(0.056)  | 0.036(0.010)  | 0.890(0.010)  |
| CAM5C_HIma     | 0.206(0.066)  | 0.042(0.012)  | 0.885(0.010)  |
| CAM5C_LIma     | 0.205(0.065)  | 0.037(0.010)  | 0.897(0.010)  |

---

## Author Comment (AC2) · 17 Oct 2020

COMMENT

The manuscript describes the uncertainty of the top-of-the-atmosphere dust direct radiative forcing due to current uncertainties in the surface soil mineralogy. Especially the importance of iron oxides is high-lighted.

The manuscript is well-organized and include detailed description of the results. It is recommended for publication after consideration of the minor comments below.

RESPONSE

Thank this reviewer much for the careful reading of the manuscript and for the com-

ments. We have made changes to the manuscript to reflect the suggestions. Text from the manuscript is quoted with double quotation marks. Please see the figure we created and cited here as Fig. R1 in the supplement material.

COMMENT

Page 2, lines 45-46: The sentence starting with "These two..." is unclear. Please rewrite.

RESPONSE

We merged this and the sentence right before into one:

"Dust aerosol (here defined as soil particles suspended in the atmosphere) perturbs the radiative energy balance directly by scattering and absorbing shortwave and longwave radiation known as the aerosol-radiation interaction (Boucher et al., 2013) and indirectly by changing the cloud albedo and lifetime by acting as cloud condensation nuclei (CCN) and ice nuclei (IN) (Nenes et al., 2014) and by increasing diabatic heating in the atmosphere and evaporating cloud (Hansen et al., 1997; Bollasina et al., 2008; Jacobson, 2012) known as the aerosol-cloud interaction (Boucher et al., 2013)."

COMMENT

Page 2, line 54: The net dust DRE is given. May you also include numbers for other DREs so the reader can have an idea about the dust DRE magnitude in relation to other processes?

RESPONSE

We now provide the SW DRE and LW DRE in addition to NET DRE.

"A recent review which synthesized data on dust abundance, optical properties, and size distribution estimated that the shortwave, longwave, and net direct radiative effects (DRE) of dust range between [-0.81, -0.15], [0.17, 0.48], and [-0.48, +0.20] W m-2, respectively (Kok et al, 2017)."

COMMENT

Page 7, line 193: Remove "in the model".

RESPONSE

Done.

COMMENT

Page 10, line 297: Remove "." after ")".

RESPONSE

Done.

COMMENT

Page 10, line 297: What is a "simple double call"?

RESPONSE

This sentence shows how to calculate the dust radiation effect in MONARCH. We rewrote this sentence as:

"The radiation flux is diagnosed twice, one with all aerosol species and the other one solely without dust aerosol to determine the DRE for bulk dust."

COMMENT

Page 15, line 465: Should $\Delta F_{upp}$ be $\Delta F_{hig}$?

RESPONSE

Yes, changed.

COMMENT

Page 21, line 650: Change "7a" to "7".

RESPONSE

Done.

COMMENT

Page 24, line 752: The region of Australia is marked with a star in Fig 11a, indicating that the result is statistically significant. This is the opposite of what is said in the text.

RESPONSE

"except Australia" deleted.

COMMENT

Page 25, line 784: Change "African" to "Africa".

RESPONSE

Done.

COMMENT

Page 26, lines 797-808: The radiative parameterization is discussed. However, it is not clear that the same radiative transfer solution method is used in both the line-by-line and the parameterized calculations. If the methods are different, for example if one of the methods use more streams, this will potentially cause significant differences.

RESPONSE

As the reviewer commented and what we had pointed out in the main text ("these GCMs underestimate the DRE and dust warming mostly due to the use of the two-stream delta-Eddington approximation in RRTMG and the radiative model's low band resolution"), the majority of the bias compared to the line-by-line calculation is due to the use of the two-streams delta-Eddington approximation in CESM over North Africa.

For clarity, we modified the text a little bit:

"these GCMs underestimate the DRE and dust warming mostly due to 1) the use of the two-stream delta-Eddington approximation (major error source) in RRTMG in comparison to the 16 streams used in the line-by-line run, and 2) the radiative model's low band resolution (minor error source compared to that in 1)".

COMMENT

Page 28, line 867: Change "that goethite" to "than goethite".

RESPONSE

Done.

COMMENT

Page 30, line 922: Change "of other" to "of the other".

RESPONSE

Done.

COMMENT

Page 31, lines 956-958: Unclear sentence (too many whiles?), please rewrite.

RESPONSE

Changed to: "For instance, CAM5 reproduced observational dust deposition within a factor of 10 in general (Fig. R1). At sites such as Colle del Lys and Colle Gnifetti in Europe, the baseline simulation in CAM5 greatly overestimated the surface deposition, while over the South Pacific the model greatly underestimated the deposition."

COMMENT

Page 31, line 958: Change "exits" to "exists".

RESPONSE

Done.

COMMENT

Page 45, line 1370: What is meant by "and are repeated"?

RESPONSE

Panel a) for example had been shown before in Fig. 1a of Scanza et al., (2015). Here we repeated the processing with the MMT taken from Clauqin et al., (1999) and show this information again. We deleted "and are repeated" as it is not a core element of the caption; we believe that solely saying "has been shown previously" is informative enough. Similarly, we also deleted "and are repeated" in the caption of Fig. 2.

COMMENT

Page 46, line 1388; page 47, line 1392; page 48, line 1404; page 52, line 1456; Fig. S13: The acronym DOD is used in the captions and figures. It is not mentioned anywhere in the text. In the text you use AOD dust. Please use only one notation for the dust AOD throughout the manuscript.

RESPONSE

Now we are using DOD all throughout the manuscript in the captions, figures, and texts.

COMMENT

Page 53, lines 1474-1475: Unclear sentence, please rewrite.

RESPONSE

Now it reads like: "All the model results were processed onto 2.5°x1.9° (longitude by latitude) horizontal grids for further calculation."

COMMENT

[Figure]

Table S2: Please provide full reference to the CESM User Guide.

RESPONSE

We inserted a link to the table caption:

http://www.cesm.ucar.edu/models/cesm1.0/cam/docs/description/cam5_desc.pdf

Please also note the supplement to this comment:
https://acp.copernicus.org/preprints/acp-2020-547/acp-2020-547-AC2-supplement.pdf
* * *
[Figure]

**Supplement:**

*Supplement of*

**Response to Referee 2**

**Longlei Li et al.**

[ll859@cornell.edu](mailto:ll859@cornell.edu)

The copyright of individual parts of the supplement might differ from the CC BY 4.0 License.

[Figure]

**Figure R1.** Comparison of simulated (the baseline case; see text for details) dust surface concentration, and deposition with observations. Also shown is correlation (both passed 95% statistically significant tests) between modeling and observations over sub-domains as indicated by texts in color. The dash lines in (b) and (d) represent 10:1 (upper left) and 1:10 (bottom right) lines.

---

## Referee Comment (RC4) · Anonymous Referee #3 · 18 Oct 2020

The manuscript discusses the direct radiative effect of dust aerosol (as defined as soil particles suspended in the atmosphere) due to source mineralogy uncertainty, focusing on the relation to the dust aerosol composition. It is well-organized. I read over the manuscript and the comments from referees #1 and #2 as well as the reply of the author. Basically, I agree with those comments and the reply from the author.

However, I do have a seeming important question. This manuscript tried to estimate the direct radiative effect of dust aerosol and only considered the mineral aerosol as it is defined. The direct radiative effect of dust aerosol happened during the long-range transport of dust aerosol. While the dust aerosol travels to thousands of kilometers away from its source area, the mineral aerosol will certainly mixes and interacts with the pollution aerosol and its chemical composition of the aerosol would be changed

and its optical properties would in turn be changed, so as its direct radiative effect. Therefore, this manuscript estimate the direct radiative effect of only mineral aerosol without considering its mixing and the interactive reaction with pollution aerosol, such an approach could be the biggest fact causing the uncertainty to the estimation of the direct radiative effect???

In 2010, Huang et al. studied the "Relation between optical and chemical properties of dust aerosol" (Huang et al, JGR-atmos. , 115, D00K16, doi:10.1029/2009JD013212). The strong heterogeneous chemical reaction on dust, and the mixing of dust with various pollutants during the long‐range/regional transport of dust plumes was observed. they found the linear relationship between optical properties and aerosol chemical composition. Soluble ions, i.e., $SO_4^{2-}$, $NO_3^-$, $NH_4^+$, and $K^+$, were the major contributors to the light extinction in fine particles, while mineral aerosol contributed more to that in coarse particles. Black carbon, as a strong light absorbing species, was found to contribute to the light extinction in both fine and coarse particles. Strong absorbing of aerosol at 439 nm was observed due to the significant proportion of iron oxides in the dust aerosol other than black carbon. The transport pathways of dust, concentrations of pollutant precursors and meteorological conditions were the main factors affecting the mixing extent of pollutants with dust.

In the references of the manuscript "Quantifying the range of the dust direct radiative effect due to source mineralogy uncertainty" by Longlei Li et al." , the paper mention above of Huang et al. (JGR-Atmos, 2010) was not cited and this manuscript totally ignored the direct radiation effect caused by the mixing and interactive reaction of dust aerosol with pollution aerosol, so as the correctness of such an estimation would be questionable!

I suggest that this manuscript should have a major revision considering the mixing and interactive reaction of dust aerosol with pollution aerosol during its long-range transport in the atmosphere.

---

## Author Comment (AC3) · 6 Dec 2020

COMMENT

For indices of refraction of Corundum (Al2O3) from the UV to thermal-IR, please see Koike et al., Icarus, 114, 203-214, 1995. The imaginary refractive index at 0.4 microns is 0.043 and at 0.5 microns is 0.0382 and at 0.6 microns is 0.0367 from ISAS (Table A1). These values are ∼1/4th those for iron oxide (e.g., which is around 0.15 at 0.5 microns), thus definitely important. The authors are correct that it will depend on concentration and whether Al2O3 can be a surrogate for Al in a mixture. These are important issues to mention.

RESPONSE

[Figure]

Many thanks to the reviewer for the comment! We provide a little bit more information about Al2O3 below to address this comment.

One of the pure crystal lattices of Al2O3 is referred to as alpha-Al2O3 or corundum. One may expect the alpha-Al2O3 or corundum in a metamorphic or plutonic environment, or in regions where heavy minerals are concentrated (communication with Dr. Konrad Kandler), but it is not as common as the minerals we considered in our manuscript. The complex refractive index (CRI) that we took from Toon and Pollack (1976) and presented in our first reply to Reviewer 1 (Fig. R1 of 'Response to Referee 1') is exactly for this type of Al2O3 (alpha-Al2O3 or corundum). As we had already thoroughly discussed in our first response to the Reviewer #1 ('Response to Referee 1'), it is impossible for alpha-Al2O3 to exert considerable influence on the dust direct radiative effect (DRE) estimate.

There is no discrepancy on the CRI of Al2O3 between Toon and Pollack (1976) and Koike et al., (1995): the CRI the Reviewer #1 was showing is not for alpha-Al2O3 or corundum but for gamma-Al2O3, a second kind of Al2O3 crystal lattice. This gamma-Al2O3 should not influence dust DRE either, simply because it is exceedingly rare in dust aerosol particles. We had stated in our first reply to the Reviewer #1 ('Response to Referee 1') that "as temperature increases, the absorption coefficient (related to the ImCRI) for the pure crystal aluminum oxide can increase by dozens of times at the visible bands". For the reference of Reviewer #1 and other readers, Toon and Pollack (1976) obtained the indices at the room temperature and/or at 1200 degree and stated their Table 2 as an upper limit. Therefore, the CRI they obtained is suitable for use and was used in our first reply ('Response to Referee 1') to demonstrate that Al2O3 has little influence on the dust DRE estimate. Unfortunately, Koike et al. (1995) did not provide the specific temperature (they just reported it as "high temperature") at which the authors made the measurement. In any way, we believe that the CRI from Koike et al. (1995) should not be used by one who is interested in DRE by dust aerosol in the Earth's atmosphere.

Based on all information we presented here and in our first response to Reviewer 1 ('Response to Referee 1') , we think that there would be no problem without mentioning aluminum oxide in our manuscript.

REFERENCE FOR THIS PART

Koike, C., Kaito, C., Yamamoto, T., Shibai, H., Kimura, S. and Suto, H.: Koike-Al2O3-1995.pdf, Icarus, 203–214, 1995.

Toon, O. B., Pollack, J. B. and Khare, B. N.: The optical constants of several atmospheric aerosol species: ammonium sulfate, aluminum oxide, and sodium chloride, J. Geophys. Res., 81(33), 5733–5748, doi:10.1029/JC081i033p05733, 1976.

---

## Author Comment (AC4) · 6 Dec 2020

COMMENT

The manuscript discusses the direct radiative effect of dust aerosol (as defined as soil particles suspended in the atmosphere) due to source mineralogy uncertainty, focusing on the relation to the dust aerosol composition. It is well-organized. I read over the manuscript and the comments from referees #1 and #2 as well as the reply of the author. Basically, I agree with those comments and the reply from the author.

RESPONSE

Thanks much for Reviewer 3's careful reading of our manuscript and for the comments.

[Figure]

COMMENT

However, I do have a seeming important question. This manuscript tried to estimate the direct radiative effect of dust aerosol and only considered the mineral aerosol as it is defined. The direct radiative effect of dust aerosol happened during the long-range transport of dust aerosol. While the dust aerosol travels to thousands of kilometers away from its source area, the mineral aerosol will certainly mixes and interacts with the pollution aerosol and its chemical composition of the aerosol would be changed and its optical properties would in turn be changed, so as its direct radiative effect. Therefore, this manuscript estimate the direct radiative effect of only mineral aerosol without considering its mixing and the interactive reaction with pollution aerosol, such an approach could be the biggest fact causing the uncertainty to the estimation of the direct radiative effect???

In 2010, Huang et al. studied the "Relation between optical and chemical properties of dust aerosol" (Huang et al, JGR-atmos. , 115, D00K16, doi:10.1029/2009JD013212). The strong heterogeneous chemical reaction on dust, and the mixing of dust with various pollutants during the long range/regional transport of dust plumes was observed. they found the linear relationship between optical properties and aerosol chemical composition. Soluble ions, i.e., $SO_4^{2-}$, $NO_3^-$, $NH_4^+$, and $K^+$, were the major contributors to the light extinction in fine particles, while mineral aerosol contributed more to that in coarse particles. Black carbon, as a strong light absorbing species, was found to contribute to the light extinction in both fine and coarse particles. Strong absorbing of aerosol at 439 nm was observed due to the significant proportion of iron oxides in the dust aerosol other than black carbon. The transport pathways of dust, concentrations of pollutant precursors and meteorological conditions were the main factors affecting the mixing extent of pollutants with dust.

In the references of the manuscript "Quantifying the range of the dust direct radiative effect due to source mineralogy uncertainty" by Longlei Li et al." , the paper mention above of Huang et al. (JGR-Atmos, 2010) was not cited and this manuscript totally

ignored the direct radiation effect caused by the mixing and interactive reaction of dust aerosol with pollution aerosol, so as the correctness of such an estimation would be questionable! I suggest that this manuscript should have a major revision considering the mixing and interactive reaction of dust aerosol with pollution aerosol during its long-range transport in the atmosphere.

RESPONSE

Thanks very much for the comments and for the careful reading of the Reviewer #3. We are taking all the above as one general comment.

It is a very good point. We actually had already briefly discussed this point in the appendix of the preprint. Those discussions in the appendix are suitable to most if not all of existing papers addressing dust direct radiative effect (DRE) using models like CAM (we have the discussion in the appendix but not in the main text, because the main text has already been very long).

As stated in the manuscript, our goal is to quantify dust DRE uncertainty due to uncertainty in the soil abundance of the minerals. So, our focus is on the sensitivity of optical properties of dust to its mineralogical composition. We compared the resultant DRE range only with that induced by some of the others, which are known important for dust DRE, and whose uncertainties are well known. The complex chemical reaction is not well included in our model. Implementation of the chemical reaction to our model is outside our scope and the scope of ACP.

Nevertheless, we hypothesize that the overall effect of the heterogeneous chemistry on dust DRE in our model should be small compared to the sensitivity to iron oxides on the following basis.

1) In CAM5/6, coarse-mode dust very well dominates over fine-mode dust in the estimated dust DRE and the perturbation to the base dust DRE. Here we take the short-wave dust DRE as an example, as the dust DRE uncertainty range we estimated at the

shortwave bands is much greater than at the longwave bands (Fig. 14 of the preprint): in the baseline, coarse-mode dust DRE is ∼-0.11 W/m2 (cited value for shortwave dust DRE at the top of the atmosphere under all-sky conditions here and elsewhere below this in the reply) versus that accumulation-mode dust DRE is ∼-0.07 W/m2 and Aitken-mode dust DRE is ∼0.00 W/m2; the perturbation to the base DRE by dust in the coarse mode is between ∼[-0.21593, 0.25807] W/m2 in CAM5 versus that in the accumulation mode is between ∼[-0.01140, 0.01985] W/m2 and in the Aitken mode is between ∼[-0.00107, 0.00047] W/m2; Considering air-pollution elements are mainly found in the fine-mode dust (Huang et al., 2010; Kandler et al., 2009) due to the large surface-to-volume ratio and long residence time of the small-sized particles in the atmosphere, the chemical reaction seems unlikely to be as influential as iron oxides in our model to dust DRE

2) The soluble coating could enhance the scattering of incoming radiation and "dust" aerosol optical depth (DOD). However, the resultant large size (primarily due to water uptake) also leads to a higher dust removal rate compared to non-aging dust, decreasing DOD. Also, we had tuned our model toward a constant DOD but not a total aerosol optical depth (AOD). So, the contribution of non-dust species to the total AOD is somewhat not much relevant

3) While East Asian dust undergoes mixing that could affect its optical properties, this is much less true for dust from most other source regions. Notably, African dust is transported toward the Caribbean (across the Atlantic Ocean) or to the Mediterranean without any substantial change in its optical properties (e.g., Denjean et al., 2015, 2016). For possible mixing in the remote regions, its impact on dust DRE would be tiny according to the points in 1) and 2).

As a response, we added two sentences in the methodology section as below (will insert Huang et al., 2010 and some others as well in the revised manuscript):

"To compare the uncertainty in the DRE from mineralogy to the other effects whose uncertainties have been well quantified, we perturb the DOD and the imaginary complex refractive index of the minerals. We do not compare the resultant DRE uncertainty due to other error sources (see the appendix), such as chemical reactions that could occur on airborne dust particles with aerosols like sulfuric acid ($H_2SO_4$), nitric acid ($HNO_3$), HCL, etc. (e.g., Li and Shao, 2009; Huang et al., 2010; Tobo et al., 2010)."

REFERENCE FOR THIS PART

Denjean, C., Caquineau, S., Desboeufs, K., Laurent, B., Maille, M., Quiñones Rosado, M., Vallejo, P., Mayol-Bracero, O. L. and Formenti, P.: Long-range transport across the Atlantic in summertime does not enhance the hygroscopicity of African mineral dust, Geophys. Res. Lett., 42(18), 7835–7843, doi:10.1002/2015GL065693, 2015.

Denjean, C., Cassola, F., Mazzino, A., Triquet, S., Chevaillier, S., Grand, N., Bourrianne, T., Momboisse, G., Sellegri, K., Schwarzenbock, A., Freney, E., Mallet, M. and Formenti, P.: Size distribution and optical properties of mineral dust aerosols transported in the western Mediterranean, Atmos. Chem. Phys., 16(2), 1081–1104, doi:10.5194/acp-16-1081-2016, 2016.

Huang, K., Zhuang, G., Lin, Y., Li, J., Sun, Y., Zhang, W. and Fu, J. S.: Relation between optical and chemical properties of dust aerosol over Beijing, China, J. Geophys. Res., 115, 1–13, doi:10.1029/2009jd013212, 2010.

Kandler, K., SchüTZ, L., Deutscher, C., Ebert, M., Hofmann, H., JäCKEL, S., Jaenicke, R., Knippertz, P., Lieke, K., Massling, A., Petzold, A., Schladitz, A., Weinzierl, B., Wiedensohler, A., Zorn, S. and Weinbruch1, S.: Size distribution, mass concentration, chemical and mineralogical composition and derived optical parameters of the boundary layer aerosol at Tinfou, Morocco, during SAMUM 2006, Tellus B Chem. Phys. Meteorol., 61(1), 32–50, doi:10.1111/j.1600-0889.2008.00385.x, 2009.

Li, W. J. and Shao, L. Y.: Observation of nitrate coatings on atmospheric mineral dust particles, Atmos. Chem. Phys., 9(6), 1863–1871, doi:10.5194/acp-9-1863-2009, 2009.

Tobo, Y., Zhang, D., Matsuki, A. and Iwasaka, Y.: Asian dust particles converted into aqueous droplets under remote marine atmospheric conditions, Proc. Natl. Acad. Sci. U. S. A., 107(42), 17905–17910, doi:10.1073/pnas.1008235107, 2010.
* * *

---

## Author Response (AR2)

Thank the reviewer and the Editor much for the comments. We revised the text accordingly as shown below.

**Referee #1**

**COMMENT**

The absorption by aluminum is not trivial as they claim; since if it is strongly absorbing in one form (e.g., gamma-Al2O3in the Koike paper), then it is probably strongly absorbing in some other forms as well, and aluminum appears in multiple forms in particles, not just alpha- or gamma-Al2O3. The authors have not explored any other form of aluminum.

**RESPONSE**

Thanks for comments from this reviewer and for those from the editor. We inserted the text as the following in Section 2.3.1:

"We consider the set of climatically important minerals identified in the soil compilations of C1999 and J2014, although other minerals may be important, especially in specific regions. However, optical analyses of aerosolized soil samples show that shortwave absorption varies most strongly with iron oxides like hematite and goethite (Moosmuller et al 2012, Di Biagio et al 2019), suggesting that other radiatively active minerals are mainly present in small concentrations."

**Referee #3**

**COMMENT**

1. The authors declared that their focus is on the sensitivity of optical properties of dust to its mineralogical composition and the implementation of the chemical reaction to the model is outside their scope and the scope of ACP. I may only accept the saying of "the focus of this manuscript is only on the relationship of the optical property to the mineralogical composition." However, the saying of "Implementation of the chemical reaction to their model is outside the scope of ACP" is totally wrong!
2. The authors asserted that "while East Asian dust undergoes mixing that could affect its optical properties, this is much less true for dust from most other source regions." This is really not true. Such a statement is an indication of the authors are not familiar with the field of the global long-range transport processes of dust aerosol.

3. Based on above 1 & 2, I would like to suggest that the authors need to have some corrections to the above statements in the related parts of this manuscript. They need to declare that the focus of this paper is only on the relationship of the optical properties of dust to its mineralogical composition. However, they need to mention that the mixing and interactions of dust with pollution aerosols should be further studied in the future.

**RESPONSE**

Thank this reviewer and the editor much for the comments.

We added the following to Section 2.3.1 of the revised manuscript:

> "We do not compare our results with the resultant DRE uncertainty due to other error sources (see the appendix), such as mixing and chemical reaction of dust with pollution aerosols (e.g., $H_2SO_4$, $HNO_3$, and HCL) (Li and Shao, 2009; Huang et al., 2010; Tobo et al., 2010), which we leave as a field of future study."

In the manuscript, we had not included the two statements quoted in (1) and (2) of this reviewer's comments, and there is no part in the article related to those statements.

We had pointed out that the focus of this study is on dust DRE uncertainty range induced by dust mineralogical composition in the introduction and in the title:

[revised manuscript text omitted]